# Parvalbumin-positive interneurons mediate neocortical-hippocampal interactions that are necessary for memory consolidation

Frances Xia[1,2], Blake A Richards[3,4], Matthew M Tran[3,4], Sheena A Josselyn[1,2,5,6], Kaori Takehara-Nishiuchi[4,5,6]*, Paul W Frankland[1,2,5,6]*

[1]Department of Physiology, University of Toronto, Toronto, Canada; [2]Program in Neurosciences and Mental Health, Hospital for Sick Children, University Avenue, Toronto, Canada; [3]Department of Biological Sciences, University of Toronto Scarborough, Toronto, Canada; [4]Department of Cell and Systems Biology, University of Toronto, Toronto, Canada; [5]Department of Psychology, University of Toronto, Toronto, Canada; [6]Institute of Medical Sciences, University of Toronto, Toronto, Canada

**\*For correspondence:**
takehara@psych.utoronto.ca (KT-N);
paul.frankland@sickkids.ca (PWF)

**Competing interests:** The authors declare that no competing interests exist.

**Abstract** Following learning, increased coupling between spindle oscillations in the medial prefrontal cortex (mPFC) and ripple oscillations in the hippocampus is thought to underlie memory consolidation. However, whether learning-induced increases in ripple-spindle coupling are necessary for successful memory consolidation has not been tested directly. In order to decouple ripple-spindle oscillations, here we chemogenetically inhibited parvalbumin-positive (PV[+]) interneurons, since their activity is important for regulating the timing of spiking activity during oscillations. We found that contextual fear conditioning increased ripple-spindle coupling in mice. However, inhibition of PV[+] cells in either CA1 or mPFC eliminated this learning-induced increase in ripple-spindle coupling without affecting ripple or spindle incidence. Consistent with the hypothesized importance of ripple-spindle coupling in memory consolidation, post-training inhibition of PV[+] cells disrupted contextual fear memory consolidation. These results indicate that successful memory consolidation requires coherent hippocampal-neocortical communication mediated by PV[+] cells.

DOI: https://doi.org/10.7554/eLife.27868.001

## Introduction

Rhythmic oscillations that occur during sleep and periods of quiet wakefulness are thought to be important for memory consolidation (*Diekelmann and Born, 2010*). Specifically, during periods of rest, hippocampal sharp-wave ripples, a form of high frequency network oscillation (100–250 Hz), are observed in temporal proximity to prefrontal cortical oscillations called spindles (12–15 Hz) (*Siapas and Wilson, 1998*). This temporal correlation, referred to as ripple-spindle coupling, is thought to support communication between the hippocampus and prefrontal cortex required for memory consolidation (*Buzsáki, 1989*, *1996*; *Clemens et al., 2011*; *Dudai et al., 2015*; *Frankland and Bontempi, 2005*; *Girardeau and Zugaro, 2011*; *Igarashi, 2015*; *Peyrache et al., 2009*; *Schwindel and McNaughton, 2011*; *Siapas and Wilson, 1998*; *Sirota et al., 2003*; *Staresina et al., 2015*; *Wierzynski et al., 2009*; *Wilson and McNaughton, 1994*). Consistent with this hypothesis, cortical electrical stimulation both enhances ripple-spindle coupling and improves performance on an object-location task (*Maingret et al., 2016*). However, whether increased ripple-

**eLife digest** Sleep contributes to the strengthening of memories. During non-dreaming sleep, neurons in regions of the brain that form and store memories – such as the hippocampus and prefrontal cortex – fire in rhythmic waves. The neurons in the hippocampus tend to fire during a wave that repeats up to 250 times per second, called sharp-wave ripples. Meanwhile, in the prefrontal cortex, the neurons tend to fire during a lower frequency wave that repeats 12 to 15 times per second, called spindles.

During sleep and quiet wakefulness, hippocampal ripples often synchronize with prefrontal spindles; that is, both waves tend to occur at approximately the same time. Many neuroscientists think this allows the brain regions to better communicate with one another, which in turn should help the brain to strengthen memories. Consistent with this possibility, rodents that learn a new task show more synchrony between ripples and spindles afterwards. But no one had actually tested whether this increase in ripple-spindle synchrony does strengthen the rodent's memory of the task. It was also unclear how the brain achieves such an increase.

Xia et al. suspected that this process involved a group of inhibitory brain cells called parvalbumin-positive interneurons. These cells act like timekeepers, and help to synchronize the firing of groups of neurons. Xia et al. now show that training mice to associate an environment with a mild electric shock made it more likely that the animals would show ripple-spindle synchrony. Yet, inhibiting the activity of parvalbumin-positive interneurons in either the hippocampus or prefrontal cortex blocked this effect. It also prevented sleep from strengthening the animals' memory of the link between the environment and the shock.

Patients with Alzheimer's disease have fewer parvalbumin-positive interneurons. By showing that these neurons help strengthen new memories, these findings may explain why losing them can impair memory. Restoring or replacing interneuron activity could be a promising therapeutic avenue to explore.

DOI: https://doi.org/10.7554/eLife.27868.002

spindle coupling following learning is necessary for memory consolidation is unknown. Furthermore, the specific cell types that underlie this phenomenon have not yet been identified.

In the hippocampus, parvalbumin-positive ($PV^+$) interneurons play a key role in regulating temporal correlations in activity. More specifically, in the CA1 region of the hippocampus, $PV^+$ cells are not required for the generation of ripple oscillations, but appear to be important for the timing of ripples and the synchronization of spiking during ripples. $PV^+$ cells exhibit phase-locked firing with ripples (*Klausberger et al., 2003*), and optogenetic inhibition of CA1 $PV^+$ cells disrupts this phase-locking (*Gan et al., 2017*) and the coherence of spiking during ripples in CA1 (*Stark et al., 2014*), without impacting the probability of ripple occurrence (*Gan et al., 2017*). Less is known about the role of $PV^+$ cells in regulating temporal correlations during oscillations in the mPFC. But, as with ripples in CA1, $PV^+$ cell activity is phase-locked to spindles in the mPFC (*Averkin et al., 2016*; *Hartwich et al., 2009*; *Peyrache et al., 2011*), suggesting a similar role of $PV^+$ cells in promoting coherent cortical population activity. The promotion of temporal coherence by $PV^+$ cells during ripples and spindles matches previous findings showing that $PV^+$ basket cells can act as a 'clocking mechanism' in circuits to ensure specific cell populations fire at appropriate times (*Freund and Katona, 2007*). Given the importance of spike-synchrony for communication between circuits (*Wang et al., 2010*), such mechanisms may be critical for inter-regional communication events such as increased ripple-spindle coupling following learning. This raises the possibility that increased ripple-spindle coupling depends on the activity of $PV^+$ cells. If so, then inhibition of $PV^+$ cell activity in either CA1 or mPFC should perturb inter-regional communication by altering ripple and spindle coherence.

To test the hypotheses that (1) $PV^+$ cells mediate increases in ripple-spindle coupling following learning, and (2) that this increase in coupling is necessary for memory consolidation, we trained mice using contextual fear conditioning. This form of learning engages plastic processes in the hippocampus, including CA1 (*Johansen et al., 2011*; *Maren et al., 2013*), and the mPFC, including the anterior cingulate cortex (ACC) (*Vetere et al., 2011*; *Zhao et al., 2005*). We used $PV^+$ cell-specific

Cre driver mice to express chemogenetic constructs allowing us to selectively inhibit PV$^+$ cells in the ACC or CA1 following training. To investigate the role of PV$^+$ cells in promoting increased ripple-spindle coupling, we performed in vivo electrophysiological recordings in mice post-training. As expected, we observed an increase in the probability of ripple-spindle coupling following contextual fear conditioning. Notably, post-training inhibition of PV$^+$ cell activity in the ACC or CA1 did not alter ripple or spindle incidence, but eliminated the learning-induced increase in ripple-spindle coupling. Consistent with this finding, inhibition of PV$^+$ cell activity in either ACC or CA1 also impaired contextual fear memory consolidation. These data indicate that PV$^+$ cells play an important role in enhancing hippocampal-neocortical dialogue following learning, and that this communication is important for memory consolidation.

## Results

### Chemogenetic inhibition of PV$^+$ cells

To target PV$^+$ interneurons in the ACC or CA1, we micro-infused an adeno-associated virus (AAV) that expresses the inhibitory Designer Receptor Exclusively Activated by Designer Drugs (DREADD) hM4Di with a fluorescent reporter (mCherry) in a Cre-recombinase-dependent manner (AAV-DIO-hM4Di-mCherry) in mice expressing Cre-recombinase only in PV$^+$ cells (PV-Cre mice) (*Armbruster et al., 2007*; *Hippenmeyer et al., 2005*; *Sohal et al., 2009*). Four weeks following surgery, numerous mCherry$^+$/PV$^+$ interneurons were observed in the ACC or CA1, respectively (*Figure 1a*; *Figure 1—figure supplement 1a*; *Figure 1—figure supplement 2*). Over 85% of endogenous PV$^+$ cells were mCherry$^+$, reflecting efficient infection rates (*Figure 1b*, n = 10). Moreover, >93% of mCherry$^+$ cells expressed PV, indicating that infection was limited to the target cell type (*Figure 1c*, n = 10) (*Sohal et al., 2009*).

DREADDs are activated by the synthetic ligand, clozapine-N-oxide (CNO). To verify that CNO-induced activation of hM4Di suppresses PV$^+$ interneuron activity, we used whole-cell patch clamp to record from ACC slices from PV-Cre mice infected with the DREADD viral vector, AAV-DIO-hM4Di-mCherry. To further control for any off-target effects of CNO, or any effects caused by the metabolic conversion of CNO to clozapine (*Gomez et al., 2017*), we also performed the same experiments using the control vector, AAV-DIO-mCherry (*Figure 1d*; hM4Di-mCherry$^+$ n = 12, hM4Di-mCherry$^-$ n=10, mCherry$^+$ n = 13, mixed-model permutation test, 1000 permutations, [hM4Di-mCherry$^+$ versus hM4Di-mCherry$^-$ versus mCherry$^+$]: p=0.001). mCherry$^+$ cells from both hM4Di- and control vector-infused mice exhibited much higher spiking rates than mCherry$^-$ cells across all current levels tested prior to CNO application, verifying that infection was limited to fast-spiking PV$^+$ interneurons (*Klausberger et al., 2003*). CNO induced hyperpolarization of hM4Di-infected PV$^+$ cells, as bath application of CNO decreased firing rates of hM4Di-mCherry$^+$, but not mCherry$^-$, or mCherry$^+$ cells in mice micro-infused with the control vector (*Figure 1e*; mixed-model permutation test, 1000 permutations, [hM4Di-mCherry$^+$ versus hM4Di-mCherry$^-$ versus mCherry$^+$] x [pre-CNO versus post-CNO]: p=0.001; individual cell firing rates pre- and post-CNO are shown in *Figure 1—figure supplement 3*). Furthermore, CNO decreased the input resistance of hM4Di-mCherry$^+$ cells only (*Figure 1f*; −80 pA current injection, two-way ANOVA, [hM4Di-mCherry$^+$ versus hM4Di-mCherry$^-$ versus mCherry$^+$] x [pre-CNO versus post-CNO]: $F_{32,1}$ = 13.14, p=6.8×10$^{-5}$, post hoc paired $t$-test with Bonferroni correction hM4Di-mCherry$^+$ [pre-CNO versus post-CNO], $t_{11}$ = 4.9, p=0.001, hM4Di-mCherry$^-$ [pre-CNO versus post-CNO], $t_9$ = −2.3, p=0.12, mCherry$^+$ [pre-CNO versus post-CNO], $t_{12}$ = 0.67, p=1.0), consistent with the interpretation that activation of hM4Di opens inwardly-rectifying K$^+$ channels. There were no changes in the excitability of mCherry$^-$ cells following bath application of CNO. This is likely because pyramidal cells in ex vivo slices do not receive inhibitory input from PV$^+$ cells at baseline, and therefore inhibiting PV$^+$ cells with bath application of CNO has no further effect on pyramidal cell excitability. These experiments also demonstrate that the effect of our manipulation (i.e., CNO-mediated inhibition) is specific for hM4Di$^+$ cells.

### Inhibition of PV$^+$ cells in either ACC or CA1 does not alter ripple or spindle incidence

Ripple-spindle coupling was previously found to increase following training in an odor-reward task (*Mölle et al., 2009*). Here, we tested whether coupling is similarly increased following training in an

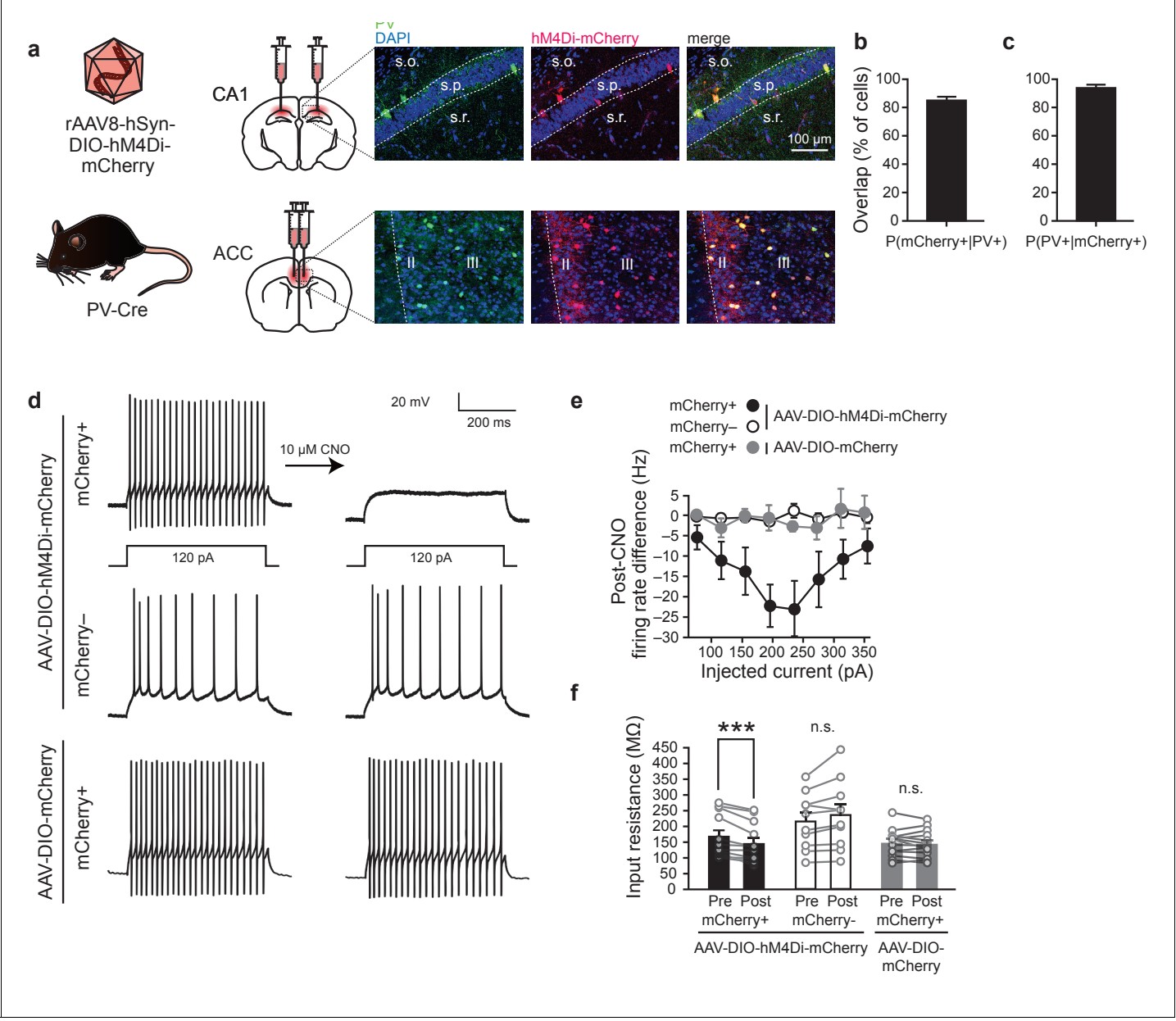

**Figure 1.** Chemogenetic inhibition of PV[+] cells. (**a**) Representative images showing co-localization of hM4Di-mCherry[+] and PV[+] cells in PV-Cre mice infused with AAV-DIO-hM4Di-mCherry virus in CA1 or ACC. (**b**) High overlap of PV[+] cells that are mCherry[+] (*n* = 10). (**c**) High overlap of mCherry[+] cells that are PV[+] cells (*n* = 10). (**d**) Representative current clamp traces in hM4Di-mCherry[+] cells and mCherry[−] cells in AAV-DIO-hM4Di-mCherry-infused mice, and mCherry[+] cells in AAV-DIO-mCherry-infused mice before and after bath application of CNO (hM4Di-mCherry[+] *n* = 12, hM4Di-mCherry[−]*n*=10, mCherry[+] *n* = 13, mixed-model permutation test, 1000 permutations, [hM4Di-mCherry[+] versus hM4Di-mCherry[−] versus mCherry[+]]: p=0.001). (**e,f**) Bath application of CNO (**e**) decreases firing rate (post-CNO − pre-CNO) in hM4Di-mCherry[+] cells (but not mCherry[−] cells, or mCherry[+] cells in AAV-DIO-mCherry-infused mice), (mixed-model permutation test, 1000 permutations, [hM4Di-mCherry[+] versus hM4Di-mCherry[−] versus mCherry[+]] x [pre-CNO versus post-CNO]: p=0.001), and (**f**) decreases input resistance in hM4Di-mCherry[+] cells (but not mCherry[−] cells, or mCherry[+] cells in AAV-DIO-mCherry-infused mice), (−80 pA current injection, two-way ANOVA, [hM4Di-mCherry[+] versus hM4Di-mCherry[−] versus mCherry[+]] x [pre-CNO versus post-CNO]: $F_{32,1}$ = 13.14, p=6.8×10[−5], post hoc paired *t*-test with Bonferroni correction hM4Di-mCherry[+] [pre-CNO versus post-CNO], $t_{11}$ = 4.9, p=0.001, hM4Di-mCherry[−] [pre-CNO versus post-CNO], $t_9$ = −2.3, p=0.12, mCherry[+] [pre-CNO versus post-CNO], $t_{12}$ = 0.67, p=1.0). Data are mean ±s.e.m., or individual mouse. (***p<0.001, n.s.: not significant).

DOI: https://doi.org/10.7554/eLife.27868.003

The following figure supplements are available for figure 1:

**Figure supplement 1.** Representative hM4Di-mCherry expression and LFP electrode locations in PV-Cre mice.

DOI: https://doi.org/10.7554/eLife.27868.004

*Figure 1 continued on next page*

*Figure 1 continued*

**Figure supplement 2.** Representative spread of hM4Di-mCherry infection in ACC and CA1.
DOI: https://doi.org/10.7554/eLife.27868.005

**Figure supplement 3.** Decrease in firing rate was observed Post-CNO in (a) hM4Di-mCherry$^+$cells ($n$ = 12), but not in (b) hM4Di-mCherry$^-$ cells ($n$ = 10), or (c) mCherry$^+$cells ($n$ = 13).
DOI: https://doi.org/10.7554/eLife.27868.006

aversively-motivated task, contextual fear conditioning (*Kim and Fanselow, 1992*). We micro-infused the AAV-DIO-hM4Di-mCherry vector in either the ACC or CA1 of PV-Cre mice, and recorded local field potentials (LFPs) in both regions to simultaneously detect spindles and ripples (*Figure 1—figure supplement 1b*). Mice were trained in contextual fear conditioning and immediately following training administered either CNO or vehicle. ACC and CA1 activity was recorded both pre-training (one day before training) and post-training (*Figure 2a*). Because ripple-spindle coupling is observed most commonly during sleep, we measured ripples (100–250 Hz) and spindles (12–15 Hz) during non-REM (NREM) periods in the pre- and post-training recording sessions using previously established criteria (*Boyce et al., 2016*; *Klausberger et al., 2003*; *Phillips et al., 2012*) (*Figure 2b*). Inhibiting PV$^+$ cells in either the ACC or CA1 with CNO did not alter the incidence of ripples (*Figure 2c*; Virus-ACC: $n$ = 8 per group; two-way repeated measures ANOVA pre-training versus post-training x Vehicle (Veh) versus CNO; pre-training versus post-training $F_{1,14}$ = 1.77, p=0.20; Veh versus CNO $F_{1,14}$ = 0.0007, p=0.98; interaction $F_{1,14}$ = 2.91, p=0.11; Virus-CA1: $n$ = 8 per group; pre-training versus post-training $F_{1,14}$ = 1.317, p=0.27; Veh versus CNO $F_{1,14}$ = 3.63, p=0.077; interaction $F_{1,14}$ = 0.10, p=0.76), consistent with previous reports using genetic manipulation of PV$^+$ cells (*Gan et al., 2017*; *Rácz et al., 2009*). This finding contrasts with a previous study in which inhibiting CA3 PV$^+$ cells disrupted ripple generation (*Schlingloff et al., 2014*), and suggests that PV$^+$ cells may play region-specific roles in modulating ripple oscillations. CNO-mediated inhibition of PV$^+$ cells in either the ACC or CA1 did not alter the incidence of spindles (*Figure 2d*; Virus-ACC: $n$ = 8 per group; pre-training versus post-training $F_{1,14}$ = 1.48, p=0.24; Veh versus CNO $F_{1,14}$ = 2.25, p=0.16; interaction $F_{1,14}$ = 3.54, p=0.081; Virus-CA1: $n$ = 8 per group; pre-training versus post-training $F_{1,14}$ = 0.039, p=0.85; Veh versus CNO $F_{1,14}$ = 0.002, p=0.96; interaction $F_{1,14}$ = 2.74, p=0.12). Furthermore, CNO did not affect ripple or spindle amplitude (*Figure 2—figure supplement 1a–b*), induce seizure-like activity (i.e., high frequency oscillations) (*Figure 2—figure supplement 1c–d*), nor alter sleep architecture (total NREM, NREM epoch duration) (*Figure 2—figure supplement 1e–f*).

## Inhibition of PV$^+$ cells in either ACC or CA1 eliminates learning-induced increases in ripple- spindle coupling

Having established that CNO-induced inhibition of PV$^+$ cells does not alter ripple or spindle incidence, we next asked whether inhibition of PV$^+$ cells affects the co-incidence of these two oscillations. We computed the cross-correlation between ripple and spindle amplitudes and observed a conditioning-dependent increase in ripple-spindle coupling in vehicle-treated mice. CNO-induced inhibition of PV$^+$ cells post-training eliminated the conditioning-dependent increase in coupling (*Figure 3*; *Figure 3b*: ACC: top; $n$ = 8 per group; pre-training versus post-training $F_{1,14}$ = 2.88, p=0.11; Veh versus CNO $F_{1,14}$ = 0.15, p=0.70; interaction $F_{1,14}$ = 6.68, p=0.022; *post hoc* Bonferroni's test, Veh pre-training versus Veh post-training p=0.018, CNO pre-training versus CNO post-training p>0.999; CA1: bottom; $n$ = 8 per group; pre-training versus post-training $F_{1,14}$ = 0.46, p=0.51; Veh versus CNO $F_{1,14}$ = 0.09, p=0.77; interaction $F_{1,14}$ = 8.42, p=0.012; *post hoc* Bonferroni's test, Veh pre-training versus Veh post-training p=0.048, CNO pre-training versus CNO post-training p=0.28; *Figure 3c*: ACC: top; $n$ = 8 per group; Welch's $t$-test $t_{9.24}$ = 2.46, p=0.035; Veh versus one one-sample $t$-test $t_7$ = 2.59, p=0.036; CNO versus one one-sample $t$-test $t_7$ = 0.17, p=0.87; CA1: bottom; Pre-training-normalized peak correlation coefficients, $n$ = 8 per group; Mann-Whitney p=0.015; Veh versus one one-sample Wilcoxon signed rank test, p=0.008; CNO versus one one-sample Wilcoxon signed rank test, p=0.31). An identical pattern was observed using other measures of coupling (cross-correlation of ripple and spindle events [*Figure 2—figure supplement 1g–h*] and ripple-spindle joint occurrence rate [*Figure 2—figure supplement 1i*]). The peak levels of ripple-spindle coupling, during both Pre- and Post-training, were significantly higher than chance in all ACC- and CA1-infused mice (an example is shown in *Figure 3—figure supplement 1a*). This suggests that the

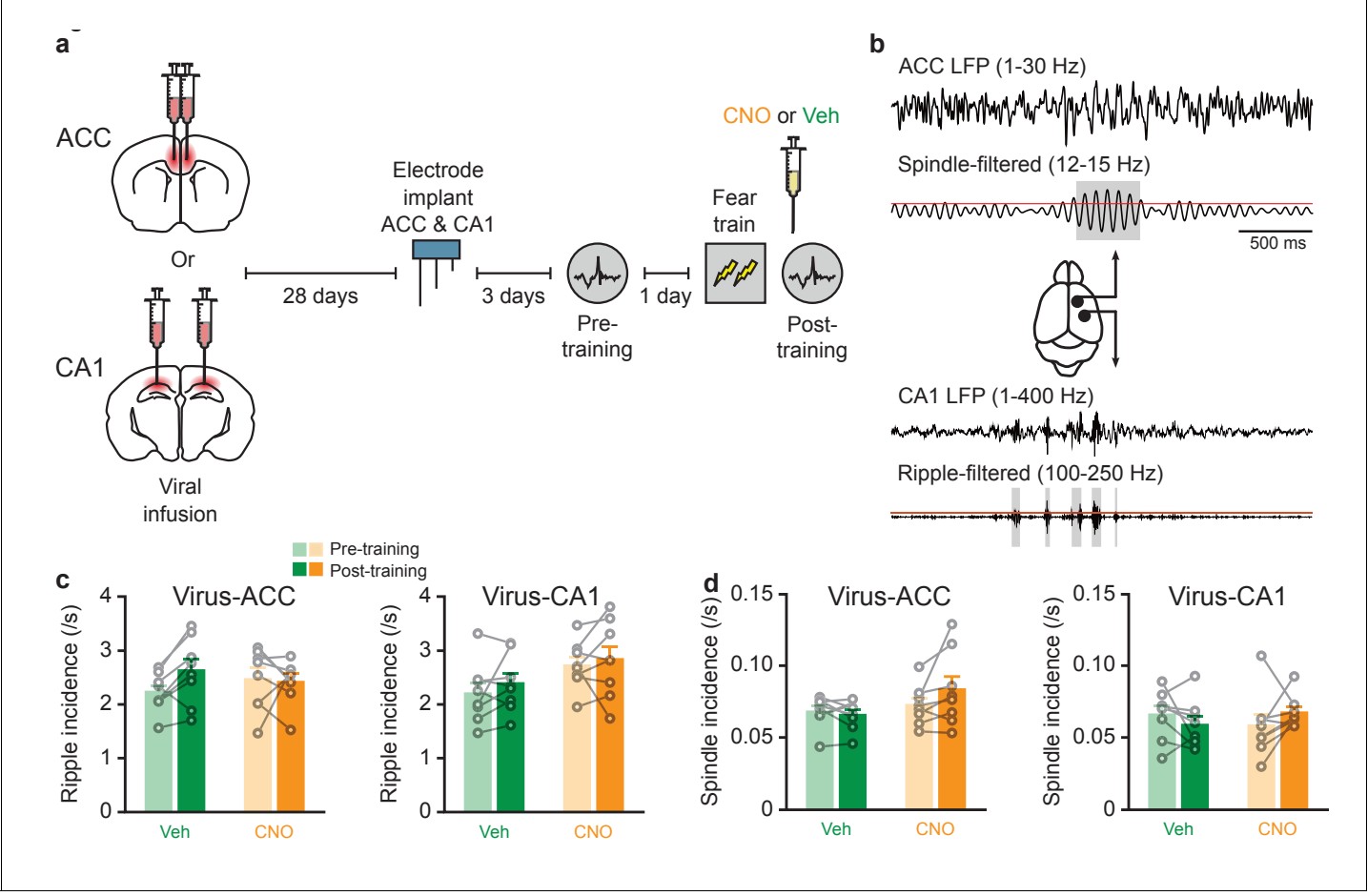

**Figure 2.** Inhibition of PV⁺cell in ACC or CA1 does not alter ripple or spindle incidence. (**a**) Experimental design. (**b**) Example traces of LFPs recorded in ACC (top two traces, low-pass filtered, and spindle-band filtered) and CA1 (bottom two traces, low-pass filtered, and ripple-band filtered), during a typical sleep session in one animal. Grey regions indicate spindles (top) and ripples (bottom) detected in ACC and CA1 LFPs, respectively. Red lines denote amplitude threshold used. Grey boxes denote ripple or spindle windows that passed detection threshold. (**c,d**) No change (**c**) in ripple incidence in mice micro-infused with virus in ACC ($n$ = 8 per group; two-way repeated measures ANOVA pre-training versus post-training x Vehicle (Veh) versus CNO; pre-training versus post-training $F_{1,14}$ = 1.77, p=0.20; Veh versus CNO $F_{1,14}$ = 0.0007, p=0.98; interaction $F_{1,14}$ = 2.91, p=0.11) or CA1 ($n$ = 8 per group; pre-training versus post-training $F_{1,14}$ = 1.317, p=0.27; Veh versus CNO $F_{1,14}$ = 3.63, p=0.077; interaction $F_{1,14}$ = 0.10, p=0.76), or (**d**) spindle incidence in mice miroinfused with virus in ACC ($n$ = 8 per group; pre-training versus post-training $F_{1,14}$ = 1.48, p=0.24; Veh versus CNO $F_{1,14}$ = 2.25, p=0.16; interaction $F_{1,14}$ = 3.54, p=0.081) or CA1 ($n$ = 8 per group; pre-training versus post-training $F_{1,14}$ = 0.039, p=0.85; Veh versus CNO $F_{1,14}$ = 0.002, p=0.96; interaction $F_{1,14}$ = 2.74, p=0.12). Data are individual mouse, or mean ±s.e.m.

DOI: https://doi.org/10.7554/eLife.27868.007

The following figure supplement is available for figure 2:

**Figure supplement 1.** Inhibition of PV⁺cells in ACC or CA1 does not alter ripple or spindle amplitude, induce seizures, or alter sleep architecture, but impairs learning-induced increase in ripple-spindle coupling.

DOI: https://doi.org/10.7554/eLife.27868.008

baseline coupling still likely reflected a significant, continuous communication between ACC and CA1, but this level was dynamically modulated by fear learning. Importantly, CNO treatment had no effect on this conditioning-dependent increase in ripple-spindle coupling in mice micro-infused with the control vector (AAV-DIO-mCherry) into the ACC, indicating that the combination of hM4Di and CNO administration was necessary for the observed effects in vivo (***Figure 3—figure supplement 1b***). Our findings that post-conditioning inhibition of PV⁺ cells in either the ACC or CA1 eliminated ripple-spindle coupling indicates that intact PV⁺ cell activity in both regions is necessary for coordinating the enhanced hippocampal-neocortical communication following learning.

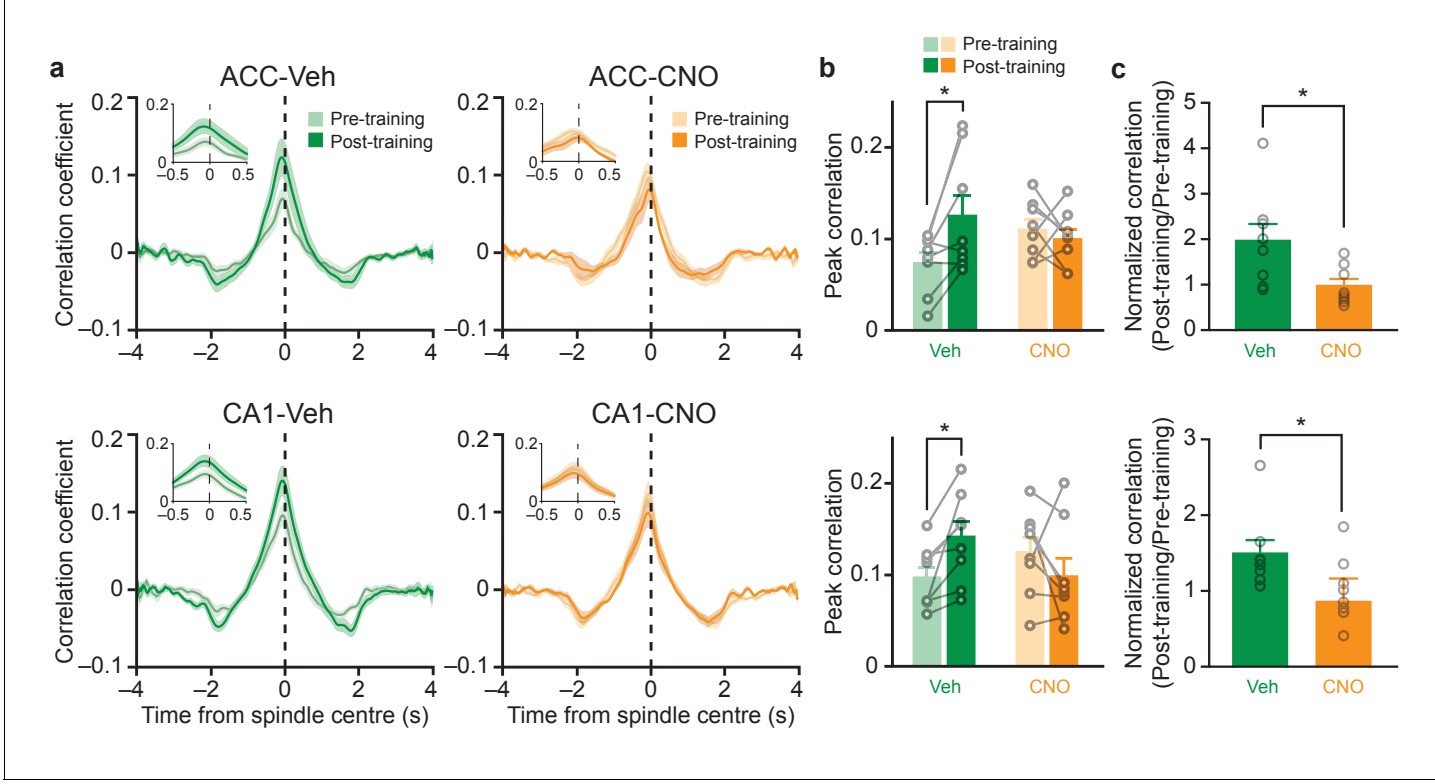

**Figure 3.** Inhibition of PV[+]cell in ACC or CA1 eliminates learning-induced increases in ripple-spindle coupling. (a) Learning-induced increases in cross-correlation between spindle and ripple amplitude in Veh-treated mice are prevented in CNO-treated mice micro-infused with hM4Di-mCherry in ACC or CA1. Insets show correlation within ±0.5 s of spindle centre. (b) Peak cross-correlation coefficients quantified from (a), in mice micro-infused with virus in ACC (top; $n = 8$ per group; pre-training versus post-training $F_{1,14} = 2.88$, p=0.11; Veh versus CNO $F_{1,14} = 0.15$, p=0.70; interaction $F_{1,14} = 6.68$, p=0.022; *post hoc* Bonferroni's test, Veh pre-training versus Veh post-training p=0.018, CNO pre-training versus CNO post-training p>0.999), or CA1 (bottom; $n = 8$ per group; pre-training versus post-training $F_{1,14} = 0.46$, p=0.51; Veh versus CNO $F_{1,14} = 0.09$, p=0.77; interaction $F_{1,14} = 8.42$, p=0.012; *post hoc* Bonferroni's test, Veh pre-training versus Veh post-training p=0.048, CNO pre-training versus CNO post-training p=0.28). (c) Pre-training-normalized peak correlation coefficients in mice micro-infused with virus in ACC ($n = 8$ per group; Welch's *t*-test $t_{9.24} = 2.46$, p=0.035; Veh versus one one-sample *t*-test $t_7 = 2.59$, p=0.036; CNO versus one one-sample *t*-test $t_7 = 0.17$, p=0.87), or CA1 (Pre-training-normalized peak correlation coefficients, $n = 8$ per group; Mann-Whitney p=0.015; Veh versus one one-sample Wilcoxon signed rank test, p=0.008; CNO versus one one-sample Wilcoxon signed rank test, p=0.31). Data are individual mouse, or mean ±s.e.m. (*p<0.05).

DOI: https://doi.org/10.7554/eLife.27868.009

The following figure supplements are available for figure 3:

**Figure supplement 1.** Probability of ripple-spindle coupling is significantly greater than chance; and learning-induced increase in ripple-spindle coupling is not prevented by CNO in mice infused with the control virus; similar to the effect on ripple-spindle coupling, inhibition of PV[+]cell in the ACC or CA1 eliminates learning-induced increases in ripple-delta coupling, without changing the time lag between baseline ripple and spindle or delta oscillations.

DOI: https://doi.org/10.7554/eLife.27868.010

**Figure supplement 2.** The learning-induced increase in ripple-spindle coupling is transient.

DOI: https://doi.org/10.7554/eLife.27868.011

We additionally examined the relationship between ripples and ACC delta oscillations since ripples are also coupled to delta oscillations (*Sirota et al., 2003*), and enhancement of cortical delta oscillations is associated with improved memory (*Marshall et al., 2006*). Similar to the effects of inhibiting PV[+] cells on disrupting ripple-spindle coupling, we observed that the post-conditioning increase in coupling between ripple and ACC delta oscillations was eliminated by inhibition of PV[+] cells in either the ACC or CA1 (*Figure 3—figure supplement 1c–d*). Importantly, inhibiting PV[+] cells did not affect the time lag between baseline ripple and spindle, or between ripple and delta, peak correlation (*Figure 3—figure supplement 1e–f*). Thus, inhibition of PV[+] cells prevents learning-induced increases in the probability of coupling of hippocampal-neocortical oscillations, but not the baseline interactions.

## Chronic post-training inhibition of PV⁺ cells in either ACC or CA1 impairs consolidation of contextual fear memory

If increased ripple-spindle coupling is essential for memory consolidation (*Igarashi, 2015*), then post-training inhibition of PV⁺ interneurons should impair memory consolidation. We first assessed whether PV⁺ interneurons were activated following learning. Analysis of the activity-regulated gene, *Fos*, shows that following fear conditioning, PV⁺ cell activity was elevated in both CA1 and ACC (compared to untrained control mice), indicating that this population of cells is strongly activated by learning (*Figure 4—figure supplement 1a*). These results are consistent with previous studies showing strong activation of inhibitory interneurons following learning (*Pi et al., 2013*; *Sparta et al., 2014*), and, specifically, PV⁺ cells following fear conditioning (*Donato et al., 2013*; *Restivo et al., 2015*; *Ruediger et al., 2011*).

To directly assess whether intact PV⁺ cell activity in the CA1 or ACC is required for memory consolidation, we trained mice in contextual fear conditioning and then administered CNO or vehicle for 4 weeks. Mice were then tested drug-free. Inhibition of PV⁺ cells in the ACC impaired consolidation of contextual fear memory, with CNO-treated mice freezing less compared to vehicle-treated controls. Similarly, chronic, post-training suppression of PV⁺ cells in CA1 impaired consolidation of contextual fear memory (*Figure 4a*; ACC: Veh $n = 6$, CNO $n = 8$, Mann-Whitney test p=0.028; CA1: Veh $n = 7$, CNO $n = 9$, $t$-test $t_{14} = 3.42$, p=0.004). Inhibiting PV⁺ interneurons in either region immediately prior to testing did not affect freezing during test (*Figure 4b*; ACC: Veh $n = 9$, CNO $n = 8$, $t$-test $t_{15} = 0.44$, p=0.66; CA1: Veh $n = 6$, CNO $n = 5$, $t$-test $t_9 = 0.28$, p=0.78), indicating that PV⁺ cell activity is not necessary for memory retrieval.

Using ex vivo patch-clamp experiments, we verified that chronic (month-long) CNO treatment inhibited hM4Di-infected neurons without altering baseline neuronal excitability (*Figure 4c–e*; *Figure 4d*: mCherry⁺ Veh $n$=14, CNO $n = 20$, mCherry⁻ Veh $n = 14$, CNO $n = 15$, mixed-model permutation test, 1000 permutations, CNO versus Veh: p=0.77; *Figure 4e*: mCherry⁺ Veh $n$=14, CNO $n = 20$, mCherry⁻ Veh $n = 14$, CNO $n = 15$, voltage clamp, mixed-model permutation test, 1000 permutations, CNO versus Veh: p=0.88). Furthermore, analysis of the activity-regulated gene, *Fos*, confirmed that CNO water treatment reduced retrieval-induced activation of hM4Di-infected neurons in both CA1 and ACC (*Figure 4f–h*, *Figure 4—figure supplement 1c*; *Figure 4g*: Veh $n = 4$, CNO $n = 5$, $t$-test $t_7 = 1.37$, p=0.21; *Figure 4h*: Veh $n = 4$, CNO $n = 5$, $t$-test $t_7 = 2.54$, p=0.039).

The ACC also modulates pain affect (*Bliss et al., 2016*). Therefore, it is possible that our PV manipulations in the ACC impact pain processing post-learning, rather than disrupting memory consolidation. To address this potential confound, we trained mice in a cued fear conditioning paradigm in which a tone was paired with a shock. This form of fear learning does not depend on either the CA1 or ACC (*Fanselow, 2010*; *Rajasethupathy et al., 2015*). In contrast to the effects observed in contextual fear conditioning, chronic CNO-induced suppression of ACC PV⁺ cell activity did not affect consolidation of tone fear conditioning (*Figure 4—figure supplement 2d*), suggesting that post-shock pain processing was not altered. Moreover, similar chronic CNO-induced suppression of ACC PV⁺ cell activity did not alter general exploratory or anxiety-related behaviours (*Figure 4—figure supplement 2a–b*).

## Inhibition of PV⁺ cells in the first but not fourth post-training week impairs consolidation of contextual fear memory

In these experiments, the activity of PV⁺ cells was chemogenetically suppressed for one month following training. However, in recording experiments, we detected increases in ripple-spindle coupling immediately following contextual fear conditioning, and not 7 or 14 days later (*Figure 3—figure supplement 2*). This suggests that increased ripple-spindle coupling may transiently contribute to memory consolidation, and, furthermore, that shorter periods of PV suppression might be sufficient to impair consolidation. To test this idea, mice were fear conditioned and tested 28 days later, as above. However, CNO was administered either during the first or last post-training week to temporally restrict inhibition of PV⁺ interneurons (*Figure 5a–b*; *Figure 5a*: ACC: Veh $n = 7$, CNO $n = 6$, Welch's $t$-test $t_{7.48} = 2.51$, p=0.038; CA1: Veh $n = 9$, CNO $n = 9$, $t$-test $t_{16} = 2.87$, p=0.011; *Figure 5b*: ACC: Veh $n = 7$, CNO $n = 7$, Mann-Whitney test p=0.90; CA1: Veh $n = 8$, CNO $n = 9$, $t$-test $t_{15} = 0.62$, p=0.55). CNO-induced suppression of PV⁺ cell activity in the ACC in the first, but not last, post-training week impaired consolidation of contextual fear memory. Similarly, post-training

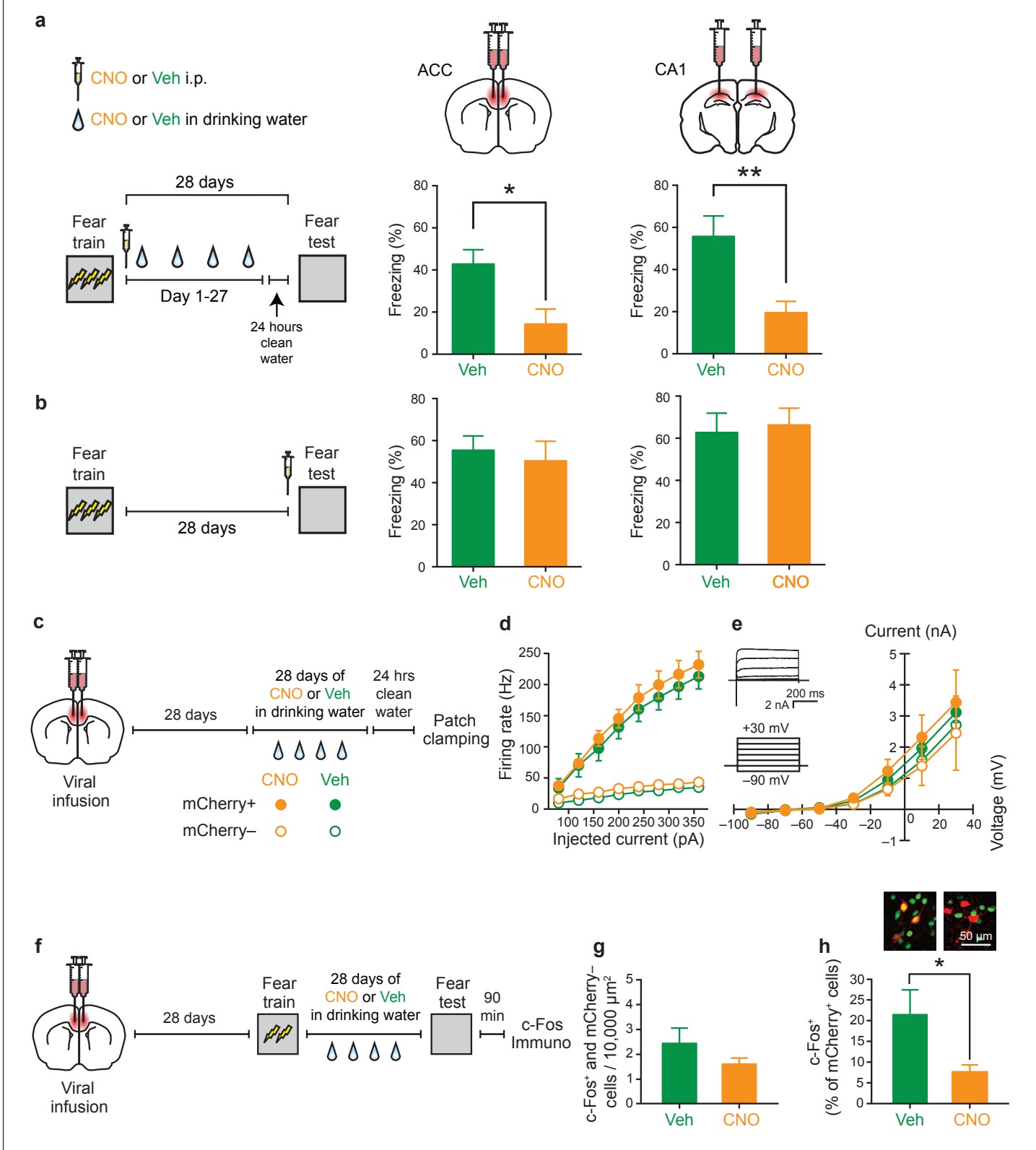

**Figure 4.** Inhibition of PV+cell in ACC or CA1 during the retention delay prevents fear memory consolidation. (**a**) Decreased freezing during fear memory test (28 d following training) in mice micro-infused with hM4Di-mCherry virus in ACC or CA1 and treated with CNO versus Veh post-training (i.p. systemic injection post-training followed by drug delivery (CNO or Veh) in water for days 1–27 and 1 d clean-water washout) (ACC: Veh $n$ = 6, CNO $n$ = 8, Mann-Whitney test p=0.028; CA1: Veh $n$ = 7, CNO $n$ = 9, $t$-test $t_{14}$ = 3.42, p=0.004). (**b**) No disruption in freezing during fear memory test (28 d

*Figure 4 continued on next page*

*Figure 4 continued*

following training) in mice micro-infused with hM4Di-mCherry virus in ACC or CA1 and treated with CNO versus Veh (i.p. injection) prior to retrieval test on the 28th day (ACC: Veh $n = 9$, CNO $n = 8$, t-test $t_{15} = 0.44$, p=0.66; CA1: Veh $n = 6$, CNO $n = 5$, t-test $t_9 = 0.28$, p=0.78). (c) Design for ex vivo experiments to assess effects of chronic CNO or Veh treatment on neuronal excitability in hM4Di-mCherry-infected and non-infected cells. (d) No effect of chronic CNO on firing rates (mCherry$^+$ Veh $n=14$, CNO $n = 20$, mCherry$^-$ Veh $n = 14$, CNO $n = 15$, mixed-model permutation test, 1000 permutations, CNO versus Veh: p=0.77), or (e) potassium currents (mCherry$^+$ Veh $n=14$, CNO $n = 20$, mCherry$^-$ Veh $n = 14$, CNO $n = 15$, voltage clamp, mixed-model permutation test, 1000 permutations, CNO versus Veh: p=0.88) in mCherry$^+$ or mCherry$^-$ cells. (f) Design for in vivo experiments to assess the effect of chronic CNO treatment on retrieval-induced neuronal activation. (g) Levels of retrieval-induced c-Fos expression in ACC mCherry$^-$ cells (number of co-localized mCherry$^-$ and c-Fos$^+$/10,000 μm$^2$) were not different between groups receiving chronic CNO versus Veh. (Veh $n = 4$, CNO $n = 5$, t-test $t_7 = 1.37$, p=0.21), but (h) CNO reduced activation of hM4Di-mCherry$^+$ neurons (number of co-localized mCherry$^+$ and c-Fos$^+$ cells/total number of mCherry$^+$ cells x 100), as expected (Veh $n = 4$, CNO $n = 5$, t-test $t_7 = 2.54$, p=0.039). Data are mean ±s.e.m. (*p<0.05, **p<0.01).
DOI: https://doi.org/10.7554/eLife.27868.012

The following figure supplements are available for figure 4:

**Figure supplement 1.** Fear learning strongly activates PV$^+$cells in both ACC and CA1; 7- or 28 day treatment of CNO reduces their activity.
DOI: https://doi.org/10.7554/eLife.27868.013

**Figure supplement 2.** Chronic inhibition of PV$^+$cells does not alter anxiety level or locomotion, or alter subsequent learning or retrieval, or affect post-shock sensitivity to pain.
DOI: https://doi.org/10.7554/eLife.27868.014

suppression of PV$^+$ interneuron activity in CA1 during the first, but not last, post-training week impaired consolidation of contextual fear memory.

Suppression of PV$^+$ interneuron activity in either the ACC or CA1 produced a similar pattern of results using a weaker conditioning protocol (*Figure 5—figure supplement 1*). More importantly, we observed the same pattern of behavioral results in mice that underwent in vivo recording (*Figure 5c*; ACC: Veh $n = 8$, CNO $n = 8$, Mann-Whitney test p=0.05; CA1: Veh $n = 8$, CNO $n = 8$, t-test $t_{14} = 2.64$, p=0.020). Furthermore, analysis of the activity-regulated gene, *Fos*, confirmed that activation of hM4Di-infected neurons was reduced by week-long CNO treatment in both CA1 and ACC (*Figure 4—figure supplement 1b*).

The absence of effects on retrieval (*Figure 4b*), as well as at time points remote to training (*Figure 5b*), suggests that PV$^+$ interneuron suppression in the ACC or CA1 does not simply interfere with the ability of mice to freeze. Indeed, chronic pre-training suppression of PV$^+$ interneurons does not alter subsequent learning or retrieval (*Figure 4—figure supplement 2c*). Together, these results indicate that the increase in ripple-spindle coupling within a relatively narrow time window following training is required for successful memory consolidation.

## Inhibition of PV$^+$ cells immediately post-training impairs consolidation of contextual fear memory

To further narrow down the window in which PV$^+$ cell activity in ACC and CA1 contributes to memory consolidation, we conducted an additional set of experiments. In these experiments, mice were fear conditioned and tested 1 day later. Immediately following training, mice received a single injection of CNO or Veh (*Figure 6a*). Inhibition of PV$^+$ cells in CA1 impaired consolidation of contextual fear memory (Veh $n = 7$, CNO $n = 10$, t-test $t_{15} = 2.75$, p=0.015), consistent with a recent report (*Ognjanovski et al., 2017*). Similarly, inhibition of PV$^+$ cells in ACC impaired consolidation of contextual fear memory (Veh $n = 12$, CNO $n = 16$, t-test $t_{26} = 3.10$, p=0.0046). In contrast, inhibiting PV$^+$ interneurons in either region immediately prior to testing did not affect freezing during test (*Figure 6b*; ACC: Veh $n = 7$, CNO $n = 12$, t-test $t_{17} = 0.71$, p=0.48; CA1: Veh $n = 6$, CNO $n = 6$, t-test $t_{10} = 0.74$, p=0.94), indicating that PV$^+$ cell activity is not necessary for memory retrieval 24 hr following training.

## Discussion

Ripple-spindle coupling has been proposed to facilitate memory consolidation, and is increased following odor-reward learning (*Mölle et al., 2009*). Furthermore, promoting ripple-spindle coupling enhances consolidation of an object-location memory (*Maingret et al., 2016*). However, previous studies did not directly test whether this form of hippocampal-neocortical communication is

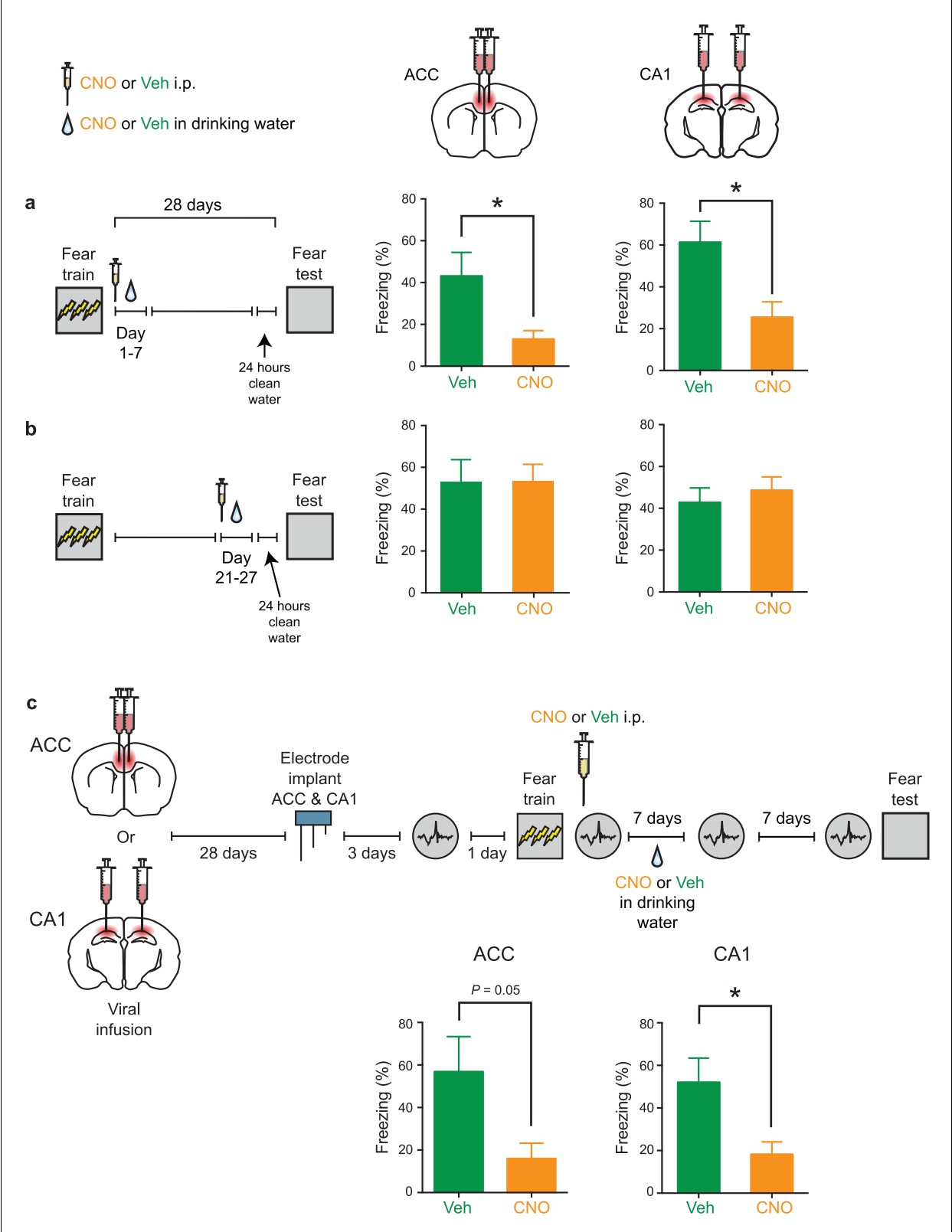

**Figure 5.** Inhibition of PV[+]cell in ACC or CA1 during first, but not fourth, post-training week prevents fear memory consolidation. (**a**) Decreased freezing during fear memory test (28 d following training) in mice micro-infused with hM4Di-mCherry virus in ACC or CA1 and treated with CNO versus Veh post-training (i.p. systemic injection post-training followed by drug delivery (CNO or Veh) in water for days 1–7) (ACC: Veh $n = 7$, CNO $n = 6$, Welch's $t$-test $t_{7.48} = 2.51$, p=0.038; CA1: Veh $n = 9$, CNO $n = 9$, $t$-test $t_{16} = 2.87$, p=0.011). (**b**) No disruption in freezing during fear memory test (28 d

*Figure 5 continued*

following training) in mice micro-infused with hM4Di-mCherry virus in ACC or CA1 and treated with CNO versus Veh post-training (drug delivery (CNO or Veh) in water for days 21–27 and 1 d clean-water washout) (ACC: Veh $n = 7$, CNO $n = 7$, Mann-Whitney test p=0.90; CA1: Veh $n = 8$, CNO $n = 9$, $t$-test $t_{15} = 0.62$, p=0.55). (c) Decreased freezing during fear memory test (14 d following training) in mice micro-infused with hM4Di-mCherry virus in ACC or CA1, implanted with LFP recording electrode and treated with CNO versus Veh post-training (i.p. systemic injection post-training followed by drug delivery (CNO or Veh) in water for days 1–7) (ACC: Veh $n = 8$, CNO $n = 8$, Mann-Whitney test p=0.05; CA1: Veh $n = 8$, CNO $n = 8$, $t$-test $t_{14} = 2.64$, p=0.020). Data are mean ±s.e.m. (*p<0.05).

DOI: https://doi.org/10.7554/eLife.27868.015

The following figure supplement is available for figure 5:

**Figure supplement 1.** Inhibition of PV[+]cells in the ACC or CA1 during retention delay also impairs memory consolidation using a weaker 2-shock fear conditioning protocol.

DOI: https://doi.org/10.7554/eLife.27868.016

necessary for successful memory consolidation, nor identify the cellular bases for mediating learning-dependent changes in ripple-spindle coupling. Here we found that contextual fear learning increased ripple-spindle coupling, and, furthermore, that chemogenetic inhibition of PV[+] cells in the ACC or CA1 both eliminated this learning-induced increase in ripple-spindle coupling and impaired memory consolidation.

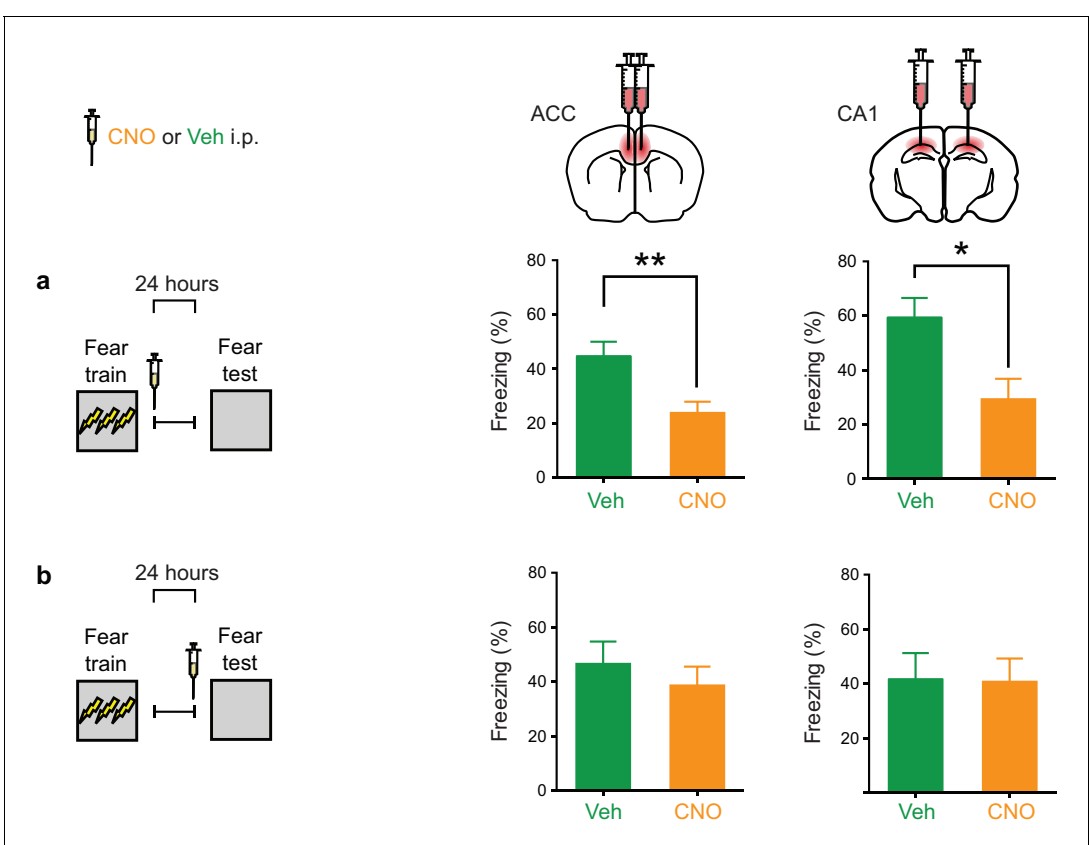

**Figure 6.** Inhibition of PV[+]cell in ACC or CA1 immediately post-training, but not during retrieval, impairs fear memory recall at 1 day. (a) Decreased freezing during fear memory test (1 d following training) in mice micro-infused with hM4Di-mCherry virus in ACC or CA1 and treated with CNO versus Veh post-training (i.p. systemic injection post-training) (ACC: Veh $n = 12$, CNO $n = 16$, $t$-test $t_{26} = 3.10$, p=0.0046; CA1: Veh $n = 7$, CNO $n = 10$, $t$-test $t_{15} = 2.75$, p=0.015). (b) No disruption in freezing during fear memory test (1 d following training) in mice micro-infused with hM4Di-mCherry virus in ACC or CA1 and treated with CNO versus Veh (i.p. injection) prior to retrieval test on the 1[st] day (ACC: Veh $n = 7$, CNO $n = 12$, $t$-test $t_{17} = 0.71$, p=0.48; CA1: Veh $n = 6$, CNO $n = 6$, $t$-test $t_{10} = 0.74$, p=0.94). Data are mean ± s.e.m. (*p<0.05, **p<0.01).

DOI: https://doi.org/10.7554/eLife.27868.017

Both mono- and multi-synaptic pathways between ACC and CA1 can support bidirectional communication between these two regions via ripple-spindle coupling. We observed an average lag between ripple and spindle peak amplitude of ~70 ms, consistent with ranges previously reported (40–244 ms; e.g., [*Peyrache et al., 2009*; *Phillips et al., 2012*; *Siapas and Wilson, 1998*; *Wang and Ikemoto, 2016*; *Wierzynski et al., 2009*]). This suggests that these two events are more likely coordinated via multiple synapses. Although the exact mechanism is unclear, there are several possibilities for bidirectional modulations. For example, ACC can modulate dorsal CA1 activity via thalamic regions, including nucleus reuniens (e.g., [*Varela et al., 2014*; *Xu and Südhof, 2013*]). Interestingly, mPFC neurons that project to the nucleus reuniens preferentially synapse onto hippocampus-projecting reuniens cells (*Vertes et al., 2007*). In addition, a subset of neurons in the nucleus reuniens project to inhibitory interneurons in CA1 (*Dolleman-Van der Weel and Witter, 2000*). Furthermore, a group of nucleus reuniens cells also has collaterals in both CA1 and mPFC, potentially coordinating activities between the two regions (*Varela et al., 2014*). CA1 can, in turn, modulate ACC via subiculum (*Varela et al., 2014*), ventral hippocampus, retrosplenial cortex (e.g.,[*Cenquizca and Swanson, 2007*]), infralimbic cortex [*Swanson, 1981*], and/or prelimbic cortex [*Thierry et al., 2000*]).

$PV^+$ cells likely coordinate ripple-spindle coupling by facilitating synchronized spiking during ripples and spindles. In CA1 and mPFC, $PV^+$ cell activity is phase-locked to ripples (*Klausberger et al., 2003*) and spindles (*Averkin et al., 2016*; *Hartwich et al., 2009*; *Peyrache et al., 2011*), respectively. In CA1, inhibition of $PV^+$ cells disrupts phase-locked firing of $PV^+$ cells to ripples, and ripple coherence (*Gan et al., 2017*; *Stark et al., 2014*). This is consistent with the proposed role of $PV^+$ cells acting as a 'clocking mechanism' in circuits, ensuring that specific cell populations fire at appropriate times (*Freund and Katona, 2007*).

Inhibition of $PV^+$ cells in the ACC or CA1 did not affect baseline probability of ripple-spindle coupling, but prevented learning-induced increases in ripple-spindle coupling. In the absence of learning, $PV^+$ cells show moderate levels of activation. However, following learning we observed strong activation of $PV^+$ cells in both regions, as well as a corresponding increase in the probability of ripple-spindle coupling. Importantly, CNO-mediated inhibition did not eliminate $PV^+$ cell activity, but reduced it to pre-learning or home cage levels (as shown in our ex vivo and in vivo experiments). Therefore, we would expect that chemogenetic inhibition of $PV^+$ cells following learning should not eliminate ripple-spindle coupling altogether, but instead, reduce it to the levels that occur in the absence of training, which is what we observed. Consistent with this idea, fear conditioning increases hippocampal network stability (*Donato et al., 2013*), and chemogenetic inhibition of $PV^+$ cells in CA1 blocks this learning-induced increase (*Ognjanovski et al., 2017*). Notably, when $PV^+$ activity levels are driven below baseline levels via other techniques, there is an associated reduction in the probability of ripple-spindle coupling, even in the absence of learning (*Phillips et al., 2012*). This suggests that the overall levels of $PV^+$ cell activity regulate the probability of ripple-spindle coupling. Accordingly, strong activation of $PV^+$ cells during learning (*Donato et al., 2013*; *Restivo et al., 2015*; *Ruediger et al., 2011*) may increase coherence both within and across brain regions. Synchronous activity, such as ripple-spindle coupling, is particularly effective at driving inter-regional communication and plasticity required for consolidation (*Fell and Axmacher, 2011*; *Igarashi, 2015*; *Wang et al., 2010*). Therefore, inhibition of $PV^+$ cell activity in either the CA1 or the mPFC likely prevented this learning-induced increase in coupling, by perturbing intra-regional synchrony of action potentials during ripples and spindles, and consequently, the coordination of inter-regional communication.

In contrast, inhibition of $PV^+$ cells in either ACC or CA1 immediately prior to testing did not affect recall (at 1 or 28 days post-training). Since overall activity in ACC and CA1 are known to be important for retrieval of contextual fear memories, these observations suggest that the activity of non-$PV^+$ cells was not affected by our PV manipulations. Consistent with this, the c-Fos levels in mCherry⁻ cells in these regions following CNO treatment were not altered.

Ripples are associated with simultaneous memory trace reactivation in the hippocampus and neocortex (*Peyrache et al., 2011*; *Peyrache et al., 2009*; *Schwindel and McNaughton, 2011*). Therefore, impaired ripple coherence following CA1 inhibition of $PV^+$ cells (*Stark et al., 2014*) likely reduced coordinated hippocampal output to the neocortex, and consequently decreased the probability of simultaneous memory trace reactivation in the neocortex. In the mPFC, memory trace reactivation is often followed by occurrence of spindles, and increased activation of local $PV^+$ cells (*Peyrache et al., 2011*). This is thought to favor the consolidation of recently modified synapses

during memory reactivation, while suppressing interfering inputs to the neocortex. Since ACC inhibition of PV$^+$ cells was sufficient to disrupt ripple-spindle coupling (without changing the overall incidence of spindles or ripples), this suggests that our manipulation interfered with the timely occurrence of spindles following ripples/memory reactivation. Therefore, inhibition of ACC PV$^+$ cells likely prevented the strengthening of synapses in the neocortex that is necessary for memory consolidation.

Our findings provide support for the idea that PV$^+$ cells are necessary for learning-associated increases in ripple-spindle coupling probability, and consequently, successful memory consolidation. Ripple-spindle coupling is also increased following odor-reward learning (*Mölle et al., 2009*), and therefore it seems plausible that the role of PV$^+$ interneurons is similar during consolidation of appetitively-motivated (as well as aversively-motivated) tasks. There are, however, alternative possibilities for why our PV manipulation resulted in consolidation deficit. For example, it is possible that the effects of inhibition of PV$^+$ cells outside of the sleep period (i.e., the ripple-spindle coupling window) could contribute to the consolidation deficits that we observed.

Moreover, inhibition of PV$^+$ cells may have increased lateral disinhibition and disrupted local circuit activity, in addition to disrupting global communication (i.e., ripple-spindle coupling). While we cannot definitively exclude this possibility, three pieces of evidence suggest that the observed consolidation deficits are mediated primarily by disruption of global communication. First, we found that inhibition of PV$^+$ cells in either ACC or CA1 immediately following training impaired memory tested 24 hr later. Activity in CA1, but not ACC, is critical for expression of contextual fear memory at this time point (*Frankland and Bontempi, 2005*). Therefore, if our manipulation of PV$^+$ cells activity only affected local activity, we would not predict the memory deficits following inhibition of ACC PV$^+$ cells. Second, inhibition of PV$^+$ cells had no effect on retrieval of contextual fear memories, tested either 24 hr or 28 days post-training, suggesting again that the overall local activity is relatively undisturbed. This reinforces the idea that our PV manipulation is distinct from other manipulations that more profoundly impact pyramidal cell activity in these regions. Third, consistent with this, we did not observe increased activation in mCherry$^-$ cells in targeted regions following inhibition of PV$^+$ interneurons. Therefore, the more plausible explanation is that the observed deficits are caused by disrupted global synchrony (i.e., ripple-spindle coupling).

We used a chemogenetic approach to manipulate PV$^+$ cell activity in ACC and CA1. One advantage of this approach is that chemogenetic-induced inhibition does not completely eliminate the activity of infected cells (e.g., compared to some forms of optogenetic silencing), and therefore is less likely to produce large-scale changes in overall circuit activity. Consistent with this, we did not observe a detectable increase in activation of mCherry$^-$ cells in either in vivo or ex vivo experiments. This may also explain why our PV manipulation did not produce broad changes in local field potential at theta (*Amilhon et al., 2015*) or gamma (*Sohal et al., 2009*) frequencies, as previously observed using optogenetic silencing of PV$^+$ cells. The absence of changes in the activity of non-infected neurons may also be related to the fact that PV$^+$ cells represent only a subpopulation of GABAergic interneurons in both ACC and CA1 (*Bezaire and Soltesz, 2013*; *Rudy et al., 2011*; *Tremblay et al., 2016*), and therefore it is plausible that non-infected cells in the circuit can still maintain homeostasis of spiking activity when the activity of PV$^+$ cells is suppressed. Moreover, reducing PV-mediated inhibition could lead to disinhibition of other inhibitory cell types (e.g., [*Lovett-Barron et al., 2012*]), thereby producing little overall change in excitation or inhibition.

In conclusion, here we showed that contextual fear learning increased the probability of ripple-spindle coupling. Inhibition of PV$^+$ cells in either ACC or CA1 eliminated this learning-induced enhancement and impaired fear memory consolidation. These data indicate that temporally correlated activities across brain regions are necessary for contextual fear memory consolidation, and our study provides evidence for an integral role for PV$^+$ cells in this process.

## Materials and methods

### Mice

All procedures were approved by the Canadian Council for Animal Care (CCAC) and the Animal Care Committees at the Hospital for Sick Children and the University of Toronto. Experiments were conducted on 8–12 week old male and female PV-Cre knock-in transgenic mice where Cre-

recombinase was targeted to the *Pvalb* locus, without disrupting endogenous PV expression (RRID: IMSR_JAX:017320). The PV-Cre mice were originally generated by Silvia Arber (*Hippenmeyer et al., 2005*), and obtained from Jackson Lab.

The mice were bred as homozygotes, weaned at 21 days, and group housed with 2–5 mice per cage in a temperature-controlled room with 12 hr light/dark cycle (light on during the day). All experiments were performed between 8 am and 12 pm. Mice were given *ad libitum* access to food and water. Mice were randomly assigned to experimental groups. The experimenter was aware of the experimental group assignment, as the same experimenter conducted the training and testing of all mice, but was blinded during behavioral assessment and cell counting experiments. Mice were excluded from analysis based on post-experimental histology: only mice with robust expression of the viral vector (hM4Di-mCherry) specifically in the targeted region were included. The spread of virus was estimated to be the following: CA1: AP −1.2 ~ −2.4 mm, ML ±0.2 ~ 3 mm, DV −1.5 ~ −2 mm; ACC: AP 1.2 ~ −0.2 mm; ML ±0.1 ~ 0.8 mm, DV −0.7 ~ −2 mm (*Figure 1—figure supplement 2*). For the in vivo electrophysiology experiments, only mice with correct electrode placements in both the ACC and CA1, as well as robust viral vector expression in the targeted region were included. Specifically, only mice where we could reliably detect sharp-wave ripples during the Pre-training recording sessions were included, to ensure that the electrodes were in CA1 cell layer. In rare cases where electrodes deteriorated prior to the completion of all experiments, and hence resulting in high noise background and no viable signals, subsequent recordings were not included in the analysis (*Figure 3—figure supplement 1g*. ACC-Veh, 2 mice).

## Viral micro-infusion

AAV8-hSyn-DIO-hM4Di-mCherry and AAV8-hSyn-DIO-mCherry viruses were obtained from UNC Vector Core (Chapel Hill, NC). In the DREADD receptor virus, AAV8-hSyn-DIO-hM4Di-mCherry, the double-floxed inverted open reading frame of hM4Di fused to mCherry can be expressed from the human synapsin (hSyn) promoter after Cre-mediated recombination. Similarly, in the control viral vector, AAV8-hSyn-DIO-mCherry, the double-floxed inverted open reading frame of the mCherry fluorescence tag can be expressed from the hSyn promoter after Cre-mediated recombination.

Four weeks prior to behaviour or electrophysiology experiments, PV-Cre mice were micro-infused bilaterally with one of these viral vectors (1.5 µl per side, 0.1 µl/min) in the ACC (+0.8 mm AP,±0.3 mm ML, −1.7 mm DV, from bregma according to *Paxinos and Franklin [2001]*) or CA1 (−1.9 mm AP, ±1.3 mm ML, - 1.5 mm DV). Similar to the previously described protocol (*Richards et al., 2014*), mice were pretreated with atropine sulphate (0.1 mg/kg, intraperitoneal), then anesthetized with chloral hydrate (400 mg/kg, intraperitoneal). Mice were then placed on a stereotaxic frame, and holes were drilled in the skull at the targeted coordinates. Viral vector was micro-infused at 0.1 µl/min via glass pipettes connected to a Hamilton microsyringe with polyethylene tubing. After micro-infusion, the glass pipette was left in the brain for another 5 min to allow sufficient time for the virus to diffuse. We have found that this infusion procedure produces high infection in the targeted region, without significant spread outside the region of interest (*Rashid et al., 2016*; *Richards et al., 2014*). Mice were then treated with analgesic (ketoprofen, 5 mg/kg, subcutaneous) and 1 ml of 0.9% saline (subcutaneous).

## Drug

Clozapine-N-oxide (CNO, kindly provided by Dr. Bryan Roth, University of North Carolina) was dissolved in dimethyl sulfoxide (DMSO) to produce a 10 mg/ml CNO stock solution. For i.p. injections, CNO stock solution was mixed with 0.9% saline, and injected at a dose of 5 mg/kg. The Vehicle (Veh) control group received equivalent amount of DMSO solution dissolved in 0.9% saline. For administration of CNO in the drinking water, preliminary experiments were first carried out to determine the amount of water a mouse consumes per day (approximately 3–5 ml of water/day). Based on the number of mice per cage, the amount of water required for 7 days was calculated for each cage, and 5 mg/kg of CNO/mouse/day was added to the water. We added sucrose (1%) to the drinking water to encourage CNO consumption. The control group received vehicle in 1% sucrose. For experiments that required more than 7 days of CNO/vehicle water, the water was changed every 7 days.

## Behavioural experiments

### Contextual fear conditioning

Four weeks after micro-infusion with hM4Di-mCherry virus in ACC or CA1, PV-Cre mice were trained in a standard contextual fear conditioning paradigm, as previously described (*Wang et al., 2009*). Mice were first habituated to the conditioning chamber for 120 s, then given 3 shocks (0.5 mA each, 60 s apart; 3-shock protocol), and remained in the chamber for another 60 s following the last shock.

For all experiments that involve chronic CNO treatment, mice were given clean drinking water for 24 hr before test on the 28th day. This washout period was designed such that mice could be tested drug-free. On the 28th day, mice were placed back into the training context for 5 min, without shock. The amount of time mice spent freezing (% freezing, with minimum bout of 2 s) was monitored with overhead cameras, and calculated using automatic scoring software FreezeFrame (Actimetrics). To investigate the robustness of the effect, the same experiments were performed using the 2-shock protocol, where mice were habituated to the chamber for 120 s, then received 2 foot shocks (0.5 mA), 60 s apart (*Figure 5—figure supplement 1*). Mice remained in the chamber for another 60 s following the final shock, and were then returned to the home cage.

To examine the effect of inhibiting PV$^+$ cells on retrieval, mice were injected i.p. with CNO or Veh 30 min prior to retrieval test (either 24 hr, or 28 days post-training). For acute inhibition experiments (*Figure 6a*), mice received a single i.p. injection of CNO or Veh immediately after training, and were tested 24 hr later.

To control for the possibility that chronic CNO impacts the ability to learn new information, mice first were micro-infused with hM4Di-mCherry virus in the ACC, then four weeks later, given 27 days of continuous CNO or vehicle water treatment. After 24 hr of clean water, mice were trained in contextual fear conditioning and memory assessed 24 hr later (*Figure 4—figure supplement 2c*).

### Tone fear conditioning

Four weeks prior to conditioning, mice were micro-infused with hM4Di-mCherry virus in the ACC (*Figure 4—figure supplement 2d*). Similar to the previously established protocol (*Rashid et al., 2016*), on the day of training, mice were habituated to the conditioning chamber (square chamber, grid floor, ethanol scent) for 120 s, then given 1 tone-shock pairing (60 s tone [2.8 kHz, 85 dB] co-terminating with 2 s foot shock at 0.7 mA). Immediately afterwards, mice were treated with i.p. systemic injection of CNO (5 mg/kg) or vehicle, followed by continuous CNO or vehicle water treatment from day 1–7 and regular water from day 7–28. On day 28, mice were tested in a novel context (round chamber, smooth floor, no ethanol scent) without shock (120 s no tone, followed by 60 s tone). The amount of time mice spent freezing during test was monitored and calculated, as described above.

### c-Fos analysis

To examine the effectiveness of chronic CNO treatment in suppressing PV$^+$ cell activity in vivo (*Figure 4*, *Figure 4—figure supplement 1b–c*), PV-Cre mice were first micro-infused with AAV-DIO-hM4Di-mCherry virus in the ACC or CA1, as described above. Four weeks after viral micro-infusion, mice were trained in contextual fear conditioning (2- or 3-shock protocol), treated with chronic CNO or vehicle in water, and tested at different delays (7 or 28 days). Ninety minutes post-test, mice were perfused, and their brains used for c-Fos staining (see below).

To examine the activity of PV$^+$ cells during learning, a group of PV-Cre mice either remained in home cage, or were trained in contextual fear conditioning (3-shock protocol) (*Figure 4—figure supplement 1a*). Ninety minutes post-training, all mice were perfused, and their brains used for c-Fos and PV staining (see below).

### Open field

To control for the possibility that chronic CNO alters anxiety levels, mice were micro-infused with hM4Di-mCherry virus in the ACC, then four weeks later, given 27 days of continuous CNO or vehicle water treatment. After 24 hr of clean water, mice were placed in the centre of an open square arena (45 cm x 45 cm x 20 cm height) and allowed to explore for 10 min (*Arruda-Carvalho et al., 2014*). The location of the mouse was tracked using an overhead camera. The amount of time a mouse spent in each of the 3 zones (1. Outer; 2. Middle; 3. Inner), as well as total distance traveled

(*Figure 4—figure supplement 2a–b*) was assessed using Limelight2 software (Actimetrics). An increase in anxiety is thought to be reflected as the mouse spending more time in the outer zone of the open field or showing decreased locomotor activity (*Archer, 1973*).

## Immunohistochemistry

Immunofluorescence staining was conducted as previously described (*Restivo et al., 2015*). Specifically, at the end of behaviour experiments, mice were transcardially perfused with 1x PBS followed by 10% paraformaldehyde. For the c-Fos experiment (*Figure 4f–g*, *Figure 4—figure supplement 1*), mice were perfused 90 min after behaviour test or training. Brains were fixed overnight at 4°C, and transferred to 30% sucrose solution for 48 hr. Brains were sectioned coronally using a cryostat (Leica CM1850), and 50 µm sections were obtained for the entire medial prefrontal cortex or hippocampus, for ACC- or CA1-infused animals, respectively.

For PV and c-Fos immunostaining, free-floating sections were blocked with PBS containing 2.5% bovine serum albumin and 0.3% Triton-X for 30 min. Afterwards, sections were incubated in PBS containing mouse monoclonal anti-PV primary antibody (1:1000 dilution; Sigma-Aldrich Cat# P3088 RRID:AB_477329) and rabbit polyclonal anti-c-Fos primary antibody (1:1000 dilution; Santa Cruz Biotechnology Cat# sc-52 RRID:AB_2106783) for 48 hr at 4°C. Sections were washed with PBS (3 times), then incubated with PBS containing goat anti-mouse ALEXA Fluor 488 (for PV, 1:500 dilution; Thermo Fisher Scientific Cat# A-11001 RRID:AB_2534069) and goat anti-rabbit ALEXA Fluor 633 (for c-Fos, 1:500 dilution, Thermo Fisher, USA Scientific Cat# A-21070 RRID:AB_2535731) secondary antibody for 2 hr at room temperature. Sections were washed with PBS, mounted on gel-coated slides, and coverslipped with Vectashield fluorescent mounting medium (Vector Laboratories). Images were obtained using a confocal laser scanning microscope (LSM 710; Zeiss) with a 20X objective.

For cell counting experiments (*Figures 1* and *4* and *Figure 4—figure supplement 1*), every second section in either ACC or CA1 was assessed for mCherry$^+$, PV$^+$ and c-Fos$^+$ cells. Approximately 4–6 sections/mouse were counted and averaged, with 3–6 mice/group. Transduction specificity (total numbers of PV$^+$ cells total numbers of mCherry$^+$ cells x 100), and efficiency (total numbers of mCherry$^+$ cells/total numbers of PV$^+$ cells x 100) were calculated. To evaluate the effectiveness of CNO in vivo, c-Fos co-localization in mCherry$^+$ cells (total numbers of c-Fos$^+$ and mCherry$^+$ co-localized cells/total numbers of mCherry$^+$ cells x 100) was calculated. To assess the activity in mCherry$^-$ cells, c-Fos$^+$ cells that are not co-localized with mCherry$^+$ cells in the region was also counted, and normalized to the area in the same section (total numbers of c-Fos$^+$ and mCherry$^-$ cells/10,000 µm$^2$). To evaluate the activity of PV$^+$ cells during learning, c-Fos co-localization in PV$^+$ cells in each region (total numbers of c-Fos$^+$ and PV$^+$ co-localized cells/total numbers of PV$^+$ cells x 100) was calculated.

## Ex vivo slice electrophysiology

PV-Cre mice were micro-infused with the DREADD receptor virus (AAV-DIO-hM4Di-mCherry) or the control vector (AAV-DIO-mCherry) in the ACC (as above). Mice were separated into two groups: (1) acute tests, to assess the excitability of ACC neurons upon direct application of CNO (*Figure 1*), or (2) chronic tests, to assess whether lasting changes arise in the excitability of neurons after 28 days of continuous CNO delivered in drinking water (*Figure 4c–e*).

For the acute group, 4 weeks following viral micro-infusion mice were anesthetized with 1.25% tribromoethanol (Avertin) and underwent cardiac perfusion with 10 mL of a chilled cutting solution (containing, in mM: 60 sucrose, 83 NaCl, 25 NaHCO$_3$, 1.25 NaH$_2$PO$_4$, 2.5 KCL, 0.5 CaCl$_2$, 6 MgCl$_2$, 20 D-glucose, 3 Na-pyruvate, 1 ascorbic acid), injected at a rate of approximately 2 mL/min. After perfusion, the brain was quickly removed and cut coronally (350 µm thickness) with a vibratome (Leica, VT1200S) in chilled cutting solution in order to obtain live, healthy slices containing the ACC. Slices were transferred to a recovery chamber comprising of a 50:50 mix of warm (34°C) cutting solution and aCSF (containing, in mM: 125 NaCl, 25 NaHCO$_3$, 1.25 NaH$_2$PO$_4$, 2.5 KCl, 1.3 CaCl$_2$, 1MgCl$_2$, 20 D-glucose, 3 Na-pyruvate, 1 ascorbic acid). Following 40–60 min of incubation, slices were transferred into a different incubation chamber with room temperature aCSF. Within the recording chamber, aCSF was heated to 32°C using an in-line heater (Warner Instruments, SF-28). Whole-cell current clamp recordings were made using glass pipettes filled with internal solution (comprising, in m): 126 K D-Gluconate, 5 KCl, 10 HEPES, 4 MgATP, 0.3 NaGTP, 10 Na-phosphocreatine). Glass capillary pipettes were pulled with a flaming brown pipette puller (Sutter, P-97) to tip

resistances between 3–8 MΩ. We determined the effects of acute CNO application by patching individual mCherry$^+$ or mCherry$^-$ cells and injecting square 500 ms current pulses into the cell (in 40 pA steps, ranging from −80 pA to 400 pA), both before and after CNO application (washing aCSF containing 10 µM CNO onto the slice for 10 min). We calculated the difference in firing rate (using the positive current injections) and input resistance (using the negative current injections) pre- and post-CNO application.

For the chronic group, 4 weeks following viral micro-infusion, mice were given either CNO or vehicle in their drinking water for 28 days. On the 29th day, mice received clean drinking water for 24 hr, to flush out the CNO in their system and allow testing in drug-free conditions. Extraction and incubation procedures followed those above. In addition to the current clamp recordings, voltage clamp recordings were obtained by clamping the voltage for 500 ms in 20 mV steps from −90 mV to +30 mV. To estimate the strength of the active, non-inactivating K$^+$ currents (which may have been altered by chronic CNO exposure) we measured the steady state current in the final 400 ms of the voltage step.

## In vivo electrophysiology

Four weeks after micro-infusion of hM4Di-mCherry or mCherry virus in the ACC or CA1 in PV-Cre mice, custom-made local field potential (LFP) electrodes were implanted in the ACC (+0.8 mm AP,±0.3 mm ML, −1.8 mm DV) and CA1 (−1.9 mm AP, ±1.3 mm ML,1 −1.7 mm DV). Similar to described above, mice were first anesthetized with 2% isoflurane and placed on a stereotaxic frame. Holes were drilled in the skull at the targeted coordinates, and virus was delivered as described above. Four weeks following viral vector micro-infusion, mice were implanted with LFP electrodes. Mice were anesthetized with 2% isoflurane and mounted onto a stereotaxic frame. Miniature stainless steel screw was placed in the cerebellum for ground, and a stripped stainless steel wire was inserted into the neck muscle for recording electromyogram (EMG) activity. Holes were drilled at the targeted coordinates, and custom made Teflon-coated stainless steel LFP electrodes (A-M Systems, Carlsborg, WA) bundled in 23–25G stainless steel cannulas were slowly lowered to the ACC (bipolar electrode with 0.3 mm distance between electrodes) and CA1 (tripolar electrode with 0.3 mm distance between electrodes), at the rate of 0.1 mm/s. LFP signals are referenced locally within the ACC or CA1. All wires were soldered to gold pins and inserted into to a plastic cap (PlasticsOne). The electrodes and cap were secured on the skull using dental cement. Mice were given ketoprofen (5 mg/kg, subcutaneous) and 1 ml 0.9% saline (subcutaneous) for 2 days following surgery. Mice were single-housed following surgery, to prevent potential fighting that could damage the cap.

Three days after surgery, mice were habituated to the recording chamber for two days (2 hr/day). The sound-attenuated chamber was dimly lit, and contained a tall Plexiglass cylinder, inside which mice were placed and allowed to sleep for the duration of the recording. All recording session were carried out during ZT 2–6, and LFP activities were recorded using the RZ-5 recording system (Tucker-Davis Technologies). Signal was amplified 1000 times, filtered between 1 and 400 Hz, and digitized at 2 kHz. On the second day of habituation, baseline (pre-training; *Figure 2a*) LFP activity was obtained. On the following day, mice were fear conditioned, similar to as described above. Immediately afterward, mice were given CNO (5 mg/kg) or vehicle i.p., and within 5–10 min, placed into the recording chamber to record the post-conditioning LFP activity (post-training, *Figure 2a* and 2 hr). We chose this specific delay (5–10 min), because data from many other groups show that neural activity in chemogenetic-infected cells is altered within 10–60 min following CNO injection (e. g., (*Alexander et al., 2009*) [*Figure 5c*]; (*Ryan et al., 2015*) [Figure S12]). For PV$^+$ cells specifically, a previous study used an identical chemogenetic-based approach to inhibit PV$^+$ cells (AAV-DIO-hM4Di in PV-Cre mice, same dose of CNO) (*Kuhlman et al., 2013*). They measured calcium transients following CNO injection, and observed a decrease in PV$^+$ cell activity, beginning 30–60 min following CNO injection. The delay we chose therefore allows us to capture the earliest onset of CNO-mediated effects on LFP activity.

Following the post-training recording session, mice were returned to the home cage, and given CNO or vehicle in drinking water for the next 7 days. The first consolidation recording session took place 7 days after fear conditioning (Con. 1, *Figure 3—figure supplement 1g–h* and 2 hr). All mice were then placed on clean drinking water for another 7 days, and at the end, the second consolidation recording session took place (Con. 2, *Figure 3—figure supplements 1g–h* and 2 hr). Mice were

then placed back into the fear training context for 4 min without shock, to examine their fear memory (*Figure 5c*).

At the end of the experiments, mice were anesthetized and electrolytic lesions (20 μA for 30 s for each electrode tip) were performed to verify the locations of electrodes. Mice were then transcardially perfused, and brains were sectioned and imaged to verify the spread of virus, similar to as described above. In addition, cresyl violet staining was performed on every other section in the ACC and dorsal CA1, to verify electrode locations (*Figure 1—figure supplement 1b*).

## Electrophysiological analysis

All analyses were performed offline using MATLAB (The MathWorks) and previously established methods as detailed below.

### Ripple, spindle, delta criteria

The detection criteria for ripples, spindles and delta waves are similar to the ones previously established (*Boyce et al., 2016*; *Eschenko et al., 2006*; *Maingret et al., 2016*; *Nakashiba et al., 2009*; *Phillips et al., 2012*), and manually verified and modified for current data set.

For ripple detection (*Boyce et al., 2016*; *Nakashiba et al., 2009*), the LFP obtained from CA1 pyramidal cell layer was first band-pass filtered (100–250 Hz), and amplitude was calculated using the Hilbert transform. Ripple windows were characterized as signals that exceed the amplitude threshold (three times the standard deviation). Signals that were less than 50 ms apart were merged.

For spindle detection (*Eschenko et al., 2006*; *Phillips et al., 2012*), the LFP obtained from ACC was band-pass filtered (12–15 Hz), and amplitude was calculated using the Hilbert transform. Spindle windows were characterized as signals that exceed the amplitude threshold (two times the standard deviation), with minimum and maximum duration of 200 and 2000 ms, respectively. Signals that are less than 100 ms apart were merged.

For delta detection (*Maingret et al., 2016*), the LFP obtained from ACC was band-pass filtered (1–4 Hz), and amplitude was calculated using the Hilbert transform. Delta windows were characterized as signals that exceed the amplitude threshold (1.5 times the standard deviation), with minimum and maximum duration of 150 and 500 ms, respectively. Signals that are less than 100 ms apart were merged.

To measure ripple and spindle density, the number of ripple or spindle events during NREM periods were calculated for each mouse, and averaged across mice in the same group (*Figure 2c–d*). To measure ripple and spindle amplitude, the peak instantaneous amplitude obtained using the Hilbert transform was extracted in each ripple or spindle window, and averaged across the number of ripple or spindle events in a recording session in each mouse. The values were then averaged across mice of the same group. There were no task differences between vehicle-treated mice in the ACC and CA1 group, so their results were combined (*Figure 2—figure supplement 1a–b*).

### Power spectrum analysis

Power estimates were computed using the Welch's method (MATLAB pwelch function) in series of 2 s bins, for the entire length of recording session for both the ACC and CA1 channels (*Nguyen et al., 2014*). The results were averaged across mice. To examine the possibility of seizures in CNO-treated mice, % total power in the CNO group for pre-training and post-training sessions was summed within five frequency bands (delta: 1–4 Hz; theta: 4–12 Hz; alpha: 12–20 Hz; beta: 20–40 Hz; gamma: 40–100 Hz), and averaged across animals (*Figure 2—figure supplement 1c–d*).

### Sleep scoring

Sleep stages (NREM/REM) were determined using adaptive theta/delta ratio (*Klausberger et al., 2003*) (threshold = 3.5 x mode) extracted from power spectrums during the periods where the mouse is immobile (*Figure 2—figure supplement 1e–f*, EMG amplitude <3 x mode for at least 10 s). Low theta/delta ratio (below threshold) is indicative of NREM periods, whereas high theta/delta ratio (above threshold) is characteristic of REM episodes. Due to the length of the recording, we are unable to reliably detect REM periods of significant duration.

## Cross-correlation analyses

The probability of ripple-spindle coupling (*Figure 3*, *Figure 3—figure supplement 1a–b*) and ripple-delta coupling (*Figure 3—figure supplement 1c–d*) were examined using cross-correlation of instantaneous amplitudes of LFP (*Adhikari et al., 2010*). This method was found to be sensitive and robust in detecting the directionality and lag between LFP signals in different brain regions and is independent of amplitude changes (*Adhikari et al., 2010*). Briefly, for ripple-spindle coupling, ripple amplitude was cross-correlated with spindle amplitude in the ±4 s time window from spindle centre, with sliding window at 0.01 s increments. The correlation time window was restricted to NREM sleep periods only. Correlation coefficient was obtained for each spindle-ripple pair, and averaged across all spindle windows for each mouse in a recording session, and averaged across mice in the same group. To assess whether the correlation levels measured were significantly above chance, we computed correlation at chance level (*Adhikari et al., 2010*). Specifically, the ripple amplitude time windows were pseudo-randomly shuffled 4–10 s with respect to spindle amplitude time windows for 100 times. The shifted amplitude windows were then cross-correlated. The process was performed for each mouse within each condition to generate the distribution of correlations at chance. The original correlation was considered significant if the peak value was higher than 99th percentile of the randomly generated cross-correlation peaks. Using this analysis, ripple-spindle cross-correlations across all conditions were significant in all mice.

Lag between ripple-spindle peak correlation and spindle centre was also calculated (*Figure 3—figure supplement 1e–f* [left panel]). A negative lag indicates a ripple lead, whereas a positive lag indicates a spindle lead. For ripple-delta coupling, ripple amplitude was cross-correlated with delta amplitude in the ±0.5 s time window from delta onset, with 0.01 s lag. Correlation coefficient was obtained for each delta-ripple pair, and averaged across all delta windows for each mouse in a recording session, and averaged across mice. Lag between ripple-delta peak correlation and delta onset was calculated (*Figure 3—figure supplement 1e–f* [right panel]). A negative lag indicates a ripple lead, whereas a positive lag indicates a delta lead.

To confirm our coupling results, we also assessed ripple-spindle coupling using a second method, by computing cross-correlation using ripple and spindle window centers as timestamps (*Siapas and Wilson, 1998*) (*Figure 2—figure supplement 1g–h*). Ripple timestamps were cross-correlated with spindle timestamps in the ±4 s time window, with sliding window at 0.1 s increments. Correlation coefficient was obtained for each mouse in a recording session, and the post-training correlation coefficient was normalized to pre-training for each mouse, and then averaged across mice in the same group.

## Ripple-spindle joint occurrence rates

As a third measure of ripple-spindle coupling, we calculated the number of ripple-spindle coupled events (*Maingret et al., 2016*), defined as ripple events that occur within ±0.25 s time window from spindle centre (*Figure 2—figure supplement 1i*). The values were normalized to the number of spindle events in the same recording session for a mouse. Then post-training joint occurrence rate was normalized to pre-training joint occurrence rate for each mouse, and then averaged across mice in the same group.

## Statistical analysis

No statistical tests were used to pre-determine sample size, but the sample sizes used are similar to those generally used within the field. Data were tested for normality and variance. If data from neither group were significantly non-normal and if variances are not significantly unequal, data were analyzed using parametric two-way repeated measures ANOVA, or two-sample Student's unpaired *t*-test. For comparisons between two groups, if the groups had significantly different variances (with $\alpha = 0.05$), Welch's *t*-test was used. For comparisons to a hypothetical mean of 1, one-sample *t*-test was used. Where appropriate, ANOVA was followed by *post hoc* pairwise comparisons with Bonferroni correction. If data were significantly non-normal (with $\alpha = 0.05$) or variances were significantly unequal, mixed-model permutation test, Kruskal-Wallis test or Mann-Whitney test (between-group comparisons), and Wilcoxon signed-rank test or Friedman test (within-group comparisons) were used accordingly. All tests were two-sided. Statistical analyses were performed using R and Graphpad Prism V6.

## Acknowledgements

We thank N Insel, JC Kim, M Morrissey, A Pourheidary, S Tanninen and J Volle for technical assistance and comments. This work was supported by Canadian Institutes of Health Research (CIHR) grants to PWF (FDN143227), SAJ (MOP74650), and Natural Sciences and Engineering Research Council of Canada (NSERC) grants to KT (RGPIN-2015–05458) and BAR (RGPIN-2014–04947). FX was supported by fellowships from NSERC and CIHR and MMT from NSERC. PWF and SAJ are senior fellows in the Child Brain and Development Program and the Brain, Mind and Consciousness programs, respectively, at the Canadian Institute for Advanced Research (CIFAR). BAR is an Associate Fellow in the Learning in Machines and Brains Program at CIFAR.

## Additional information

### Funding

| Funder | Grant reference number | Author |
|---|---|---|
| Canadian Institutes of Health Research | FDN143227 | Paul W Frankland |
| Canadian Institutes of Health Research | MOP74650 | Sheena A Josselyn |
| Natural Sciences and Engineering Research Council of Canada | RGPIN-2015-05458 | Kaori Takehara-Nishiuchi |
| Natural Sciences and Engineering Research Council of Canada | RGPIN-2014-04947 | Blake A Richards |
| Natural Sciences and Engineering Research Council of Canada | | Frances Xia Matthew M Tran |

The funders had no role in study design, data collection and interpretation, or the decision to submit the work for publication.

### Author contributions

Frances Xia, Conceptualization, Data curation, Formal analysis, Funding acquisition, Validation, Investigation, Visualization, Methodology, Writing—original draft, Project administration, Writing—review and editing; Blake A Richards, Conceptualization, Data curation, Formal analysis, Supervision, Funding acquisition, Validation, Investigation, Visualization, Methodology, Writing—original draft, Writing—review and editing; Matthew M Tran, Data curation, Formal analysis, Funding acquisition, Investigation; Sheena A Josselyn, Funding acquisition, Writing—original draft, Writing—review and editing; Kaori Takehara-Nishiuchi, Conceptualization, Data curation, Formal analysis, Supervision, Funding acquisition, Validation, Investigation, Visualization, Methodology, Writing—original draft, Project administration, Writing—review and editing; Paul W Frankland, Conceptualization, Supervision, Funding acquisition, Visualization, Methodology, Writing—original draft, Project administration, Writing—review and editing

### Author ORCIDs

Frances Xia http://orcid.org/0000-0001-7415-6620
Blake A Richards http://orcid.org/0000-0001-9662-2151
Sheena A Josselyn http://orcid.org/0000-0001-5451-489X
Kaori Takehara-Nishiuchi https://orcid.org/0000-0002-7282-7838
Paul W Frankland http://orcid.org/0000-0002-1395-3586

### Ethics

Animal experimentation: All procedures in this study were approved by the Canadian Council for Animal Care (CCAC) and the Animal Care Committees at the Hospital for Sick Children and the University of Toronto.

Decision letter and Author response
Decision letter https://doi.org/10.7554/eLife.27868.019
Author response https://doi.org/10.7554/eLife.27868.020

## Additional files

### Supplementary files
• Transparent reporting form
DOI: https://doi.org/10.7554/eLife.27868.018

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
