## [Decision Letter]

Thank you for submitting your article "Parvalbumin-positive interneurons mediate cortical-hippocampal interactions that are necessary for memory consolidation" for consideration by *eLife*. Your article has been favorably evaluated by Andrew King (Senior Editor) and three reviewers, one of whom is a member of our Board of Reviewing Editors. The following individuals involved in review of your submission have agreed to reveal their identity: Thomas J. McHugh (Reviewer #2); Matt Jones (Reviewer #3).

The reviewers have discussed the reviews with one another and the Reviewing Editor has drafted this decision to help you prepare a revised submission.

Summary:

In this study, Xi et al. expressed DREADS in PV cells of PV-Cre mice and show that reduced activity of PV interneurons results in a diminished cross-correlation between ripple activity recorded in CA1 and spindle oscillations obtained from the anterior cingulate cortex (ACC) after mice have been trained to contextual fear conditioning. The percentage of freezing also declined. Similar results were not observed in naïve, untrained mice. Moreover, PV interneuron silencing one week after training was effective in reducing learning (freezing) but not 3-4 weeks after training, indicating that PV interneurons are needed for the consolidation of memory. The work was reviewed by three reviewers and all three very much impressed by the quality of the work. Several concerns, particularly regarding the interpretation of the data, have been formulated by the reviewers which will need to be authors addressed.

Essential revisions:

First, the authors show in the initial figures that PV interneuron inhibition causes a reduced cross-correlation between spindle and ripple activity. Later, however, they swap to the observation of freezing behavior only and do not show examples of the underlying reduced cross-correlation in oscillatory activities between PFC and CA1. Please provide these data either in the main figures of the study or in the supplementary material.

Second, reviewer #2 was concerned with the interpretation of some of the data and therefore formulated a proposal for an additional behavioral experiment, as set out below.

Third, determine the spread of the viruses.

Fourth, please comment on the question of whether ripples during quiet wake and ripples during non-REM are functionally equivalent in terms of their contributions to memory consolidation.

Finally, be aware that a recent publication 'Ognjanovski N, Schaeffer S, Wu J, Mofakham S, Maruyama D, Zochowski M, Aton SJ. Parvalbumin-expressing interneurons coordinate hippocampal network dynamics required for memory consolidation. Nat Commun. 2017 Apr 6;8:15039. doi:10.1038/ncomms15039', which addresses related questions. This paper should be discussed in the manuscript.

Major reviewer comments:

1) It is unclear why an aversive-motivated task has been chosen for the study and not a task which does not require the involvement of fear, as, for example, an odor-reward task (Molle et al., 2009). The authors show at a later stage of the manuscript that PV interneuron silencing in weaker conditioning protocols also resulted in similar reduced freezing, but again this behavior includes freezing conditions. The question is, does inhibition of PV interneurons reduce freezing only after aversive-motivated tasks followed by consolidation of the fear memory, or can this finding be more generalized to also other behavioral (non-fear-related) conditions?

2) The authors show that PV interneuron inhibition causes a reduced cross-correlation between spindle and ripple activity. Later they swap to the observation of freezing behavior only and do not show examples of the underlying reduced cross-correlation in oscillatory activities between PFC and CA1. These data should be shown because they are the proposed underlying circuit mechanism.

3) Although the authors showed by using whole-cell recordings in vitro that bath application of CNO reduces the activity of PV interneurons expressing DREADDs, in vivo single unit recordings of fast-spiking interneurons, putative PV cells, would be the better proof of concept.

4) One question emerging from the study is 'how is the here proposed coupling between ripple CA1 and spindle PFC realized with a delay of ~70 ms?' As the authors mention, it must be a polysynaptic pathway and they provide some examples of possible morphological connections, such as the interfacing nucleus reunions. LFP recordings from all three brain areas would be one way of addressing this question.

5) The authors report that PV^+^ inhibition in the PFC or CA1 prevents the post-training increase in ripple-spindle coupling and impairs memory consolidation. The interpretation they prefer is one of the necessity of coupling to drive strengthening of cortical synapses which in turn underlies the consolidation of a 28d memory. The post-training physiology data is used to suggest that the local impact of inhibition is minimal with the primary deficit being the decreased coupling of the events. However, if the local interventions in the PFC and/or CA1 impacted the quality or stability of the recently encoded memory then in essence there would be nothing to consolidate, which is a deficit distinct from a loss of coupled reactivation events. While this may seem trivial or semantic, it is key to their primary conclusion. More importantly, there are experimental approaches to clarify these scenarios. Unfortunately just reporting that ripples and/or spindles are present and unchanged in frequency following CNO tells us nothing about their information content in terms of temporally precise spiking (replay). I realize these types of analyses would be impossible due to the manner in which the current data set was recorded. However there is a behavioral experiment that would strengthen their interpretation. Their model predicts that if the memory were to be tested when it was still 'hippocampal dependent' [in most system consolidation models ~1 week post-training] it should be intact even in animals receiving CNO. The authors do provide some limited 14d testing data in Figure 5 from recording mice that demonstrates deficits at this point, however this is a difficult time point to interpret. Ideally a behavioral experiment modeled on the experiments in Figure 5 should be included that tested context fear at the 7d time point in mice that receive 1 week of post-training CNO treatment. If the impact of PV manipulations is specific to ripple-spindle coupling and the impairment of systems consolidation then memory should be intact at this time. If impairments are seen in either CNO treated group then perhaps an alternative and/or additional explanation is warranted. If a 7d deficit is observed then local inhibition of the PV circuit in the PFC and/or CA1 may impair hippocampal consolidation perhaps through changing ripple participation/timing/replay immediately post-training and the 28d deficit may not necessarily be related to the decrease in event coupling.

6) It was fortunate that a 5mg/kg dose of CNO provided a 'goldilocks' level of inhibition such that no off-target effects (seizure/change in ripple occurrence or frequency/baseline c-Fos expression) were observed nor was baseline coupling reduced (as seen in the Phillips et al. paper), yet increased coupling of spindles and ripples following learning was abolished. Although the authors address this in the Discussion, I still wonder if they think this implies that baseline coupling is unrelated to memory consolidation and/or is simply reflective of chance events? This can be estimated by asking if the frequency of baseline events exceed what is expected by chance based on rate of occurrence of each type of event. Further, given that the authors also report that the learning included c-Fos expression in PV^+^ neurons is lost in the CNO treated mice, does this imply plasticity in the PV network similar to what has been suggested by Caroni and colleagues underlies the appearance of the 'new' coupled events? This should be addressed in greater detail in the Discussion.

7) How large is the extent of the spread of the viral manipulation? A rather large volume of virus was used at each injection site, however from the data and text presented it is impossible to infer the extent of the manipulation. This is important as the phenotypes on both the behavioral and physiological levels are system-wide. Even a rough estimate of the average dorsal/ventral and medial/lateral spread of the infected neurons would be useful.

8) The authors demonstrate that fear memory recall induces a significant increase in the fraction of PV^+^ neurons expressing c-Fos (Figure 4, Figure 4—figure supplement 1). However CNO administered before testing did not impair recall (Figure 4), suggesting that the activity driving c-Fos induction during recall is unrelated to behavior. While admittedly a minor point, these data seem at odds which each other and should be mentioned in the Discussion.

9) An additional analysis that would offer insights into an important issue in the field is whether ripples during quiet wake and ripples during non-REM are functionally equivalent in terms of their contributions to memory consolidation? Analyses should be included to show awake ripple densities and properties in relation to (a) learning and (b) CNO.

---

## [Author Response]

Essential revisions:First, the authors show in the initial figures that PV interneuron inhibition causes a reduced cross-correlation between spindle and ripple activity. Later, however, they swap to the observation of freezing behavior only and do not show examples of the underlying reduced cross-correlation in oscillatory activities between PFC and CA1. Please provide these data either in the main figures of the study or in the supplementary material.

We thank the reviewer for allowing us to clarify this important point. In one set of experiments, LFP recordings and behavioral analyses were conducted in the same mice, and these data are presented in Figures. 2, 3, and 5C. Importantly, these experiments show that a) fear conditioning increases the probability of ripple-spindle coupling, b) inhibition of PV^+^ interneurons eliminates this learning-induced increase in coupling, and c) elimination of this learning-induced increase in coupling impairs fear memory consolidation. This pattern of results was observed following inhibition of PV^+^ interneurons in either the ACC or CA1 region of the hippocampus.

In this set of experiments, inhibition of PV^+^ interneurons began immediately following fear conditioning and continued for 7 days. Fear memory was then assessed 14 days after training. These combined recording/behavioral experiments then served as a foundation for a series of behavioral experiments in which we more fully explored the time window in which PV^+^ interneuron inhibition impairs fear memory consolidation (e.g., length of inhibition [1 d vs. 1 week vs. 1 month] and timing of inhibition [during the 1^st^ vs. 4^th^ post-training week]). Given the large numbers of mice involved in these experiments, we did not perform additional LFP recordings in these mice. Moreover, we were interested in memory expression at remote time-points (i.e., 28 days post-training). We could not consistently maintain viable electrodes for that length of time, and therefore it was not possible to conduct parallel recordings in these extended experiments.

Second, reviewer #2 was concerned with the interpretation of some of the data and therefore formulated a proposal for an additional behavioral experiment, as set out below.

We thank the reviewer for this excellent suggestion, and have conducted the additional behavioral experiments as suggested. Specifically, we trained mice in contextual fear conditioning, and then inhibited PV^+^ interneurons immediately following training. Mice were then tested 24 hours later. Inhibition of PV^+^ interneurons in either the ACC or CA1 impaired subsequent memory consolidation (Figure 6 in the revised manuscript).

These data speak to the broader issue of whether our manipulation is affecting local vs. global activities. With respect to CA1, many studies have shown that contextual fear memory is dependent on CA1 activity at this 24 hour time point (e.g., pharmacological/optogenetic inhibition, and lesions). Therefore, the memory deficits that we observed might be related to disturbance of local activity. That being said, we also showed that the same manipulation disrupts global activity (e.g., ripple-spindle coupling). Therefore, from these CA1 results, it is difficult to dissociate the effects of our manipulation on local vs. global activities.

However, the ACC results suggest that the effects are most likely due to disturbance of global activity (i.e., ripple-spindle coupling). One day after training, contextual fear memory does not depend on ACC activity (e.g., Frankland et al. [2004]). Therefore, the memory deficit is unlikely to be due to disruption of local activity in the ACC alone. The more plausible explanation is that the observed deficits are caused by disrupted global synchrony (i.e., ripple-spindle coupling).

Furthermore, according to this account, disrupting ripple-spindle coupling should have no effect on retrieval at 24 hours or 28 days post-training (since we do not expect to observe ripple-spindle coupling while mice are awake and behaving, such as during memory retrieval tests). We therefore also examined the impact of inhibition of PV^+^ cells in either CA1 or ACC on retrieval at the 24 hour time point [Figure 6] (in addition to the 28 day data already presented in the original manuscript [Figure 4, Figure 5—figure supplement 1]) following contextual fear conditioning. In all cases, we found that inhibition of PV^+^ cells in either CA1 or ACC during retrieval at 24 hours or 28 days post-training had no effect on memory performance, suggesting that our manipulation is distinct from other manipulations that more profoundly impact pyramidal cell activity (i.e., direct manipulations of pyramidal cells).

Third, determine the spread of the viruses.

As suggested we have provided more comprehensive characterization of our viral infections, and in particular paid attention to the anterior/posterior, medial/lateral, and dorsal/ventral spread of our infections. We estimated the spread as suggested (CA1: AP -1.2 ~ -2.4 mm, ML ± 0.2 ~ 3 mm, DV -1.5 ~ -2 mm; ACC: AP 1.2 ~ -0.2 mm; ML ± 0.1 ~ 0.8 mm, DV -0.7 ~ -2 mm). This information is now included in the Materials and methods section. Schematics and additional representative examples of infections in each region are now shown in Figure 1—figure supplement 2. Importantly, we only included mice where infected neurons were confined to the target region, with negligible spread to neighboring regions. The criteria are stated in the Materials and methods section.

Fourth, please comment on the question of whether ripples during quiet wake and ripples during non-REM are functionally equivalent in terms of their contributions to memory consolidation.

As awake ripples are generally recorded in the conditioning environment, and we did not do recordings during fear conditioning, we cannot directly speak to the differences between awake ripple and NREM ripples in memory consolidation.

Finally, be aware that a recent publication 'Ognjanovski N, Schaeffer S, Wu J, Mofakham S, Maruyama D, Zochowski M, Aton SJ. Parvalbumin-expressing interneurons coordinate hippocampal network dynamics required for memory consolidation. Nat Commun. 2017 Apr 6;8:15039. doi:10.1038/ncomms15039', which addresses related questions. This paper should be discussed in the manuscript.

We have added discussion of the Ognjanovski et al. (2017) paper in the revised manuscript, as suggested. Our results are consistent with those of Ognjanovski et al. insofar as finding that post-training inhibition of CA1 PV^+^ interneurons impaired memory consolidation (when tested 24 hours after training). Ognjanovski et al. also showed that inhibition of PV^+^ interneurons disrupted learning-induced increases in functional connectivity and stability within CA1, resulting in reduced coherent spiking activities, bolstering the idea that PV^+^ interneurons are critical for spike timing activity in CA1. These findings are also consistent with our hypothesis that disrupted timing in CA1 can subsequently disrupt coherent communication with other brain regions (such as ACC), and therefore lead to reductions in learning-induced increases in ripple-spindle coupling.

Major reviewer comments:1) It is unclear why an aversive-motivated task has been chosen for the study and not a task which does not require the involvement of fear, as, for example, an odor-reward task (Molle et al., 2009). The authors show at a later stage of the manuscript that PV interneuron silencing in weaker conditioning protocols also resulted in similar reduced freezing, but again this behavior includes freezing conditions. The question is, does inhibition of PV interneurons reduce freezing only after aversive-motivated tasks followed by consolidation of the fear memory, or can this finding be more generalized to also other behavioral (non-fear-related) conditions?

Molle et al. (2009) found that odor-reward learning was associated with increased ripple-spindle coupling. This inspired us to examine whether the same is observed following learning in an aversively motivated task, and further explore the mechanisms underlying the learning-induced increases in ripple-spindle coupling. Although these authors did not directly manipulate PV^+^ cells in their study, it seems plausible that the role of PV^+^ interneurons would be similar during consolidation of appetitively motivated vs. aversively motivated tasks (although this remains to be tested). We make note of this in the revised Discussion (seventh paragraph).

2) The authors show that PV interneuron inhibition causes a reduced cross-correlation between spindle and ripple activity. Later they swap to the observation of freezing behavior only and do not show examples of the underlying reduced cross-correlation in oscillatory activities between PFC and CA1. These data should be shown because they are the proposed underlying circuit mechanism.

Please see above Essential revisions point 1.

3) Although the authors showed by using whole-cell recordings in vitro that bath application of CNO reduces the activity of PV interneurons expressing DREADDs, in vivo single unit recordings of fast-spiking interneurons, putative PV cells, would be the better proof of concept.

We presented several convergent lines of evidence that show that PV^+^ cell activity can be efficiently inhibited by CNO (including ex vivo patch clamp recordings, and in vivo immediate early gene analyses). We agree that in vivo single unit recordings would add to this proof of concept. However, unfortunately we were not set up to do in vivo single unit recordings in mice.

4) One question emerging from the study is 'how is the here proposed coupling between ripple CA1 and spindle PFC realized with a delay of ~70 ms?' As the authors mention, it must be a polysynaptic pathway and they provide some examples of possible morphological connections, such as the interfacing nucleus reunions. LFP recordings from all three brain areas would be one way of addressing this question.

We very much agree with this, and this is an avenue that we will explore in future work.

*5) The authors report that* PV^+^*inhibition in the PFC or CA1 prevents the post-training increase in ripple-spindle coupling and impairs memory consolidation. The interpretation they prefer is one of the necessity of coupling to drive strengthening of cortical synapses which in turn underlies the consolidation of a 28d memory. The post-training physiology data is used to suggest that the local impact of inhibition is minimal with the primary deficit being the decreased coupling of the events. However, if the local interventions in the PFC and/or CA1 impacted the quality or stability of the recently encoded memory then in essence there would be nothing to consolidate, which is a deficit distinct from a loss of coupled reactivation events. While this may seem trivial or semantic, it is key to their primary conclusion. More importantly, there are experimental approaches to clarify these scenarios. Unfortunately just reporting that ripples and/or spindles are present and unchanged in frequency following CNO tells us nothing about their information content in terms of temporally precise spiking (replay). I realize these types of analyses would be impossible due to the manner in which the current data set was recorded. However there is a behavioral experiment that would strengthen their interpretation. Their model predicts that if the memory were to be tested when it was still 'hippocampal dependent' [in most system consolidation models ~1 week post-training] it should be intact even in animals receiving CNO. The authors do provide some limited 14d testing data in Figure 5 from recording mice that demonstrates deficits at this point, however this is a difficult time point to interpret. Ideally a behavioral experiment modeled on the experiments in Figure 5 should be included that tested context fear at the 7d time point in mice that receive 1 week of post-training CNO treatment. If the impact of PV manipulations is specific to ripple-spindle coupling and the impairment of systems consolidation then memory should be intact at this time. If impairments are seen in either CNO treated group then perhaps an alternative and/or additional explanation is warranted. If a 7d deficit is observed then local inhibition of the PV circuit in the PFC and/or CA1 may impair hippocampal consolidation perhaps through changing ripple participation/timing/replay immediately post-training and the 28d deficit may not necessarily be related to the decrease in event coupling.*

Please see above Essential revisions point 2.

*6) It was fortunate that a 5mg/kg dose of CNO provided a 'goldilocks' level of inhibition such that no off-target effects (seizure/change in ripple occurrence or frequency/baseline c-Fos expression) were observed nor was baseline coupling reduced (as seen in the Phillips et al. paper), yet increased coupling of spindles and ripples following learning was abolished. Although the authors address this in the Discussion, I still wonder if they think this implies that baseline coupling is unrelated to memory consolidation and/or is simply reflective of chance events? This can be estimated by asking if the frequency of baseline events exceed what is expected by chance based on rate of occurrence of each type of event. Further, given that the authors also report that the learning included c-Fos expression in* PV^+^*neurons is lost in the CNO treated mice, does this imply plasticity in the PV network similar to what has been suggested by Caroni and colleagues underlies the appearance of the 'new' coupled events? This should be addressed in greater detail in the Discussion.*

We thank the reviewer for this suggestion. We have calculated the probability of ripple-spindle coupling at chance level for each condition in each mouse (virus in ACC vs. CA1, Pre-training vs. Post-training), and found that all correlations, including those at baseline (Pre-training), were significantly higher than chance for all mice (an example is shown in Figure 3—figure supplement 1, and the results have been added to the revised manuscript). The results are consistent with previous studies (e.g., Siapas and Wilson [1998]). This suggests that the baseline correlation still likely reflects a significant, continuous communication between the two brain regions. However, the level of coupling is dynamically modulated by learning (i.e., fear learning increases the probability of coupling). This likely reflects an increased need for cross-regional communication during consolidation, which requires increased activity of PV^+^ cells in both ACC and CA1.

We agree that our findings are consistent with those by Caroni and colleagues, and suggest that learning activates PV^+^ cells and this may lead to increased network stability (e.g., by switching the network to a high-PV-configuration and reducing synaptic turnover). This may be one of the mechanisms underlying learning-induced increases in local/global coherence. We have added this idea to the Discussion (fourth paragraph).

7) How large is the extent of the spread of the viral manipulation? A rather large volume of virus was used at each injection site, however from the data and text presented it is impossible to infer the extent of the manipulation. This is important as the phenotypes on both the behavioral and physiological levels are system-wide. Even a rough estimate of the average dorsal/ventral and medial/lateral spread of the infected neurons would be useful.

Please see above Essential revisions point 3.

*8) The authors demonstrate that fear memory recall induces a significant increase in the fraction of* PV^+^*neurons expressing c-Fos (Figure 4, Figure 4—figure supplement 1). However CNO administered before testing did not impair recall (Figure 4), suggesting that the activity driving c-Fos induction during recall is unrelated to behavior. While admittedly a minor point, these data seem at odds which each other and should be mentioned in the Discussion.*

We have now added a brief discussion on this point (Discussion, fifth paragraph), as suggested. While there was an increase in c-Fos in the PV^+^ cell populations in ACC and CA1 during tests (and a corresponding decrease in the CNO groups), we did not observe changes in c-Fos levels in mCherry^-^ following CNO treatment (compared to vehicle-treated mice; [Figure 4, Figure 4—figure supplement 1 bottom panel]). This suggests that the overall activity of non-PV^+^ cells in these two brain regions was not significantly impacted by our manipulation, and therefore we would not expect an impact on memory retrieval. This also reinforces the idea that our PV manipulation is distinct from other manipulations that more profoundly impact pyramidal cell activity in these regions.

9) An additional analysis that would offer insights into an important issue in the field is whether ripples during quiet wake and ripples during non-REM are functionally equivalent in terms of their contributions to memory consolidation? Analyses should be included to show awake ripple densities and properties in relation to (a) learning and (b) CNO.

Please see above Essential revisions point 4.